# A Genetic Algorithm for Navigating Synthesizable Molecular Spaces

**Alston Lo, Connor W. Coley, Wojciech Matusik**
Massachusetts Institute of Technology
{alston,wojciech}@csail.mit.edu, ccoley@mit.edu

## Abstract

Inspired by the effectiveness of genetic algorithms and the importance of synthesizability in molecular design, we present SynGA, a simple genetic algorithm that operates directly over synthesis routes. Our method features custom crossover and mutation operators that explicitly constrain it to synthesizable molecular space. By modifying the fitness function, we demonstrate the effectiveness of SynGA on a variety of design tasks, including synthesizable analog search and sample-efficient property optimization, for both 2D and 3D objectives. Furthermore, by coupling SynGA with a machine learning-based filter that focuses the building block set, we boost SynGA to state-of-the-art performance. For property optimization, this manifests as a model-based variant SynGBO, which employs SynGA and block filtering in the inner loop of Bayesian optimization. Since SynGA is lightweight and enforces synthesizability by construction, our hope is that SynGA can not only serve as a strong standalone baseline but also as a versatile module that can be incorporated into larger synthesis-aware workflows in the future. Our code is available at https://github.com/alstonlo/synga.

## 1 Introduction

The design of novel molecules is a costly and time-intensive endeavor, so significant effort has gone into developing computational tools to de-risk and accelerate the process. Molecular design involves a constrained optimization problem that is made challenging by the discrete combinatorial nature of molecular space and the need for sample-efficiency. The rapid development of machine learning (ML) has led to exciting advances for in silico design, with methods such as variational autoencoders (Gómez-Bombarelli et al., 2018), reinforcement learning (Olivecrona et al., 2017), GFlowNets (Bengio et al., 2021), and large language models (M. Bran et al., 2024) being proposed, to name a few. Yet among them, genetic algorithms (GAs) (Holland, 1992), a classical approach, have remained competitive for their simplicity, sample-efficiency, and exploratory power (Tripp & Hernández-Lobato, 2023; Gao et al., 2022a). This is in contrast to standard ML methods which tend to be data-hungry and struggle to extrapolate from their training sets. GAs have been used to design organic emitters (Nigam et al., 2024), polymers (Kim et al., 2021), catalysts (Seumer & Jensen, 2024), and drugs (Terfloth & Gasteiger, 2001). However, unlike ML models, classical GAs cannot learn insights from data and are reliant on expert-designed genetic operators. Thus, there is increasing interest in enhancing GAs with ML (or vice versa) (Kneiding & Balcells, 2024). This is primarily done by (1) using the GA as a subroutine within some broader ML workflow, or (2) augmenting a part of the GA (e.g., crossover) with ML (Section 2). Such work reaffirms the strength of GAs in chemistry and shows that GAs and ML can in fact be coupled synergistically.

Search power, however, matters only if domain constraints are obeyed. Many molecular generative models are synthesis-agnostic, which can lead to them proposing unstable or unsynthesizable designs (Gao & Coley, 2020). This presents a major barrier for adopting these models in real-world applications, regardless of their performance on benchmarks. While using retrosynthesis models post-hoc can alleviate this issue, they may also incur prohibitive runtime, taking minutes per evaluation (Saigiridharan et al., 2024). Instead, another promising strategy is to incorporate synthesis considerations directly into the model itself (Stanley & Segler, 2023). One class of synthesis-aware models are those that operate directly on synthesis routes, often defined in terms of a fixed catalog of purchasable building blocks and expert-defined reaction templates. These template-based models are appealing

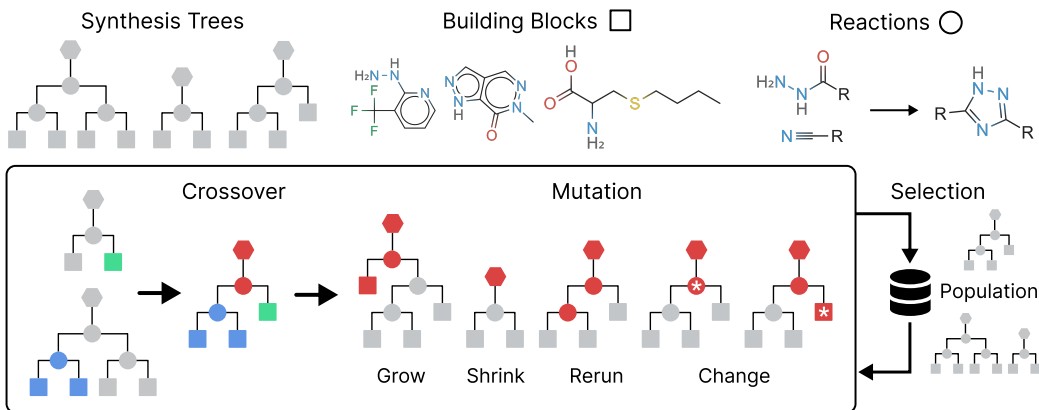

Figure 1: A graphical overview of SynGA, which operates over synthesis trees built from building blocks (squares) and reaction templates (circles). Example blocks and a reaction are drawn above using SmilesDrawer (Probst & Reymond, 2018).

because they are explicitly constrained and the molecules produced by them come automatically with plausible synthesis routes.

Given the power of GAs and the importance of synthesis constraints, a natural step is then to consider synthesis-constrained GAs. Prior work (Gao et al., 2022b; 2024; Sun et al., 2025b) has done so by augmenting an unconstrained GA with an ML model trained to project arbitrary molecules back onto synthesis space. While these methods are successful in synthesis planning and molecular design, they come with the upfront cost of training the ML model and the recurring cost of making inference calls to it. Moreover, they are reliant on the quality and generalization of the projection module, which can be difficult to train since it has to compress a combinatorially-large synthesis space. In this work, we take an alternate approach and directly embed synthesis constraints within the GA itself through custom genetic operators. More precisely, our contributions are:

1. A GA, SynGA, that evolves synthesis routes directly (Figure 1) and is thereby explicitly synthesis-constrained. SynGA is simple and ML-free, which makes it a nice baseline and subroutine for future algorithms, similar to unconstrained GAs.

2. An elegant way to enhance SynGA through ML-guided building block filtering, wherein a lightweight model is trained to dynamically restrict the block set depending on the optimization task. For property optimization, this leads to SynGBO, a Bayesian optimization algorithm that uses SynGA and block filtering in its inner loop.

3. Extensive benchmarks of SynGA on various optimization tasks, where we show that SynGA or its augmented versions achieve state-of-the-art performance. These include synthesizable analog search and sample-efficient property optimization, for 2D and 3D objectives.

## 2 BACKGROUND

**Synthesis-aware molecular design.** Synthesizability can be incentivized through heuristics (Ertl & Schuffenhauer, 2009), reward design (Guo & Schwaller, 2025), or fragmentation schemes (Polishchuk, 2020; Lewell et al., 1998; Degen et al., 2008), but it can also be enforced by engineering the generative process itself. In this last case, a key design choice lies in how reactions are formalized. Template-based methods (Gao et al., 2022b; Button et al., 2019) use a library of expert-defined reaction rules, whereas template-free methods (Wang et al., 2022; Bradshaw et al., 2019; 2020) use an ML model to predict products. While both approaches have limitations and neither *guarantee* synthesizability, an advantage of templates is they induce well-defined search spaces that do not rely on a black-box predictor. Recent template-based synthesis-aware algorithms include evolutionary algorithms (Vinkers et al., 2003; Button et al., 2019; Wang et al., 2025), tree search (Swanson et al., 2024), projection models (Gao et al., 2022b; Luo et al., 2024; Gao et al., 2024; Sun et al., 2025b), GFlowNets (Seo et al., 2025; Koziarski et al., 2024; Cretu et al., 2025), flow matching (Shen et al., 2025), reinforcement learning (Gao et al., 2024; Gottipati et al., 2020; Horwood & Noutahi,

2020), and large language models (Sun et al., 2025a; Wang et al., 2025). By constraining generation, these methods yield molecules with routes that can be plausibly executed using standard laboratory protocols. For a more comprehensive review, we refer readers to Stanley & Segler (2023).

**Genetic algorithms.** GAs can be defined on a variety of molecular representations such as SMILES (Yoshikawa et al., 2018), SELFIES (Nigam et al., 2022; 2020; Krenn et al., 2020; Lo et al., 2023), molecular graphs (Jensen, 2019; Tripp & Hernández-Lobato, 2023; Fu et al., 2022; Yoshikawa et al., 2018), and synthesis routes (Gao et al., 2022b; 2024; Sun et al., 2025b). Although GAs sometimes surpass ML baselines (Tripp & Hernández-Lobato, 2023; Gao et al., 2022a), the strongest results now emerge when they are paired with ML. Such hybridization can manifest in multiple roles: guiding a neural apprentice policy (Ahn et al., 2020); boosting exploitation or exploration in GFlowNets (Kim et al., 2024), Augmented Memory (Guo & Schwaller, 2024), and retrieval-augmented generation (Lee et al., 2024); and optimizing acquisition functions in Bayesian optimization (Tripp et al., 2021; Tripp & Hernández-Lobato, 2024). When combined with synthesizable projection models, GAs can also navigate synthesizable space efficiently (Gao et al., 2022b; 2024; Sun et al., 2025b). Other works incorporate ML *inside* the GA itself, through ML-guided genetic operators (Kim et al., 2025; Fu et al., 2022) or adding learned selection pressures (Nigam et al., 2020; 2022), highlighting the rich design space for future hybrid methods.

## 3 APPROACH

We are interested in the space of the synthesizable molecules that can be obtained from a given set of building blocks and reactions. Formally, if $\mathcal{M}$ is the universe of molecules, let $\mathcal{B} \subseteq \mathcal{M}$ be a finite subset of purchasable building blocks. In addition, let $\mathcal{R}$ be a finite set of reaction rules, each one being a function $R: \{S \in 2^{\mathcal{M}} \mid |S| = \mathrm{arity}(R)\} \to 2^{\mathcal{M}}$ that maps a (multi)set of $\mathrm{arity}(R) \leq 2$ reactants to a set of possible products, assuming each reaction is unary or binary. In practice, $\mathcal{R}$ is implemented by expert-defined SMARTS strings (templates), which we make invariant to input order by applying them to every permutation of the reactants and taking the union of the products as the final output. A reaction may return no products (i.e., $\varnothing$), if the input reactants are incompatible. It may also return multiple products, for example, by design or due to ambiguous regioselectivity.

New synthesizable molecules can be formed by iteratively applying reactions to the building blocks. A synthesis route producing a molecule $M$ can be represented as an unordered binary tree, where each node $v$ is labeled with a molecule $M_v$ and a reaction $R_v$ such that $M_{\mathrm{root}} = M$ and:

1. If $v$ is a leaf node, then $M_v \in \mathcal{B}$.

2. If $v$ is an internal node, then $M_v \in R_v(\{M_w \mid w \text{ is a child of } v\})$ and, implicitly, $v$ has exactly $\mathrm{arity}(R_v)$ children and they correspond to compatible reactants of $R_v$. Assigning an $M_v$ here is necessary for disambiguation, since $R_v$ can yield multiple products.

Conversely, any tree satisfying (1) and (2) can be interpreted as a valid synthesis route yielding a synthesizable molecule. Hence, there is a surjection $\mathcal{T} \twoheadrightarrow \mathcal{M}_S$ from the set of synthesis trees $\mathcal{T}$ to the space of synthesizable molecules $\mathcal{M}_S \subseteq \mathcal{M}$, and we can cast search problems over $\mathcal{M}_S$ as ones over $\mathcal{T}$, which admits more compact and structured representations for ML. Here, we show that it is both possible and effective to directly search over $\mathcal{T}$ using SynGA, a simple genetic algorithm defined on synthesis trees.

### 3.1 GENETIC ALGORITHMS

Inspired by natural selection, genetic algorithms (GAs) (Holland, 1992) are a class of optimization algorithm that have been shown to be powerful at navigating chemical space. Generally, GAs work by iteratively updating a population of individuals through genetic operators that bias it towards higher fitness (by which we mean the value under the objective function). Commonly, GAs define three types of genetic operators: (1) crossover, which hybridizes two individuals to form a new one, (2) mutation, which locally perturbs the result of crossover, and (3) selection. At each step or generation, pairs of parents are sampled from the population and crossover and mutation are applied to produce offspring. Subsets of the offspring and current population are carried into the next generation as the new population. The selection operator defines the parent sampling and population update rules,

often in a manner that favors fitter individuals. Algorithm 1 gives the general structure of SynGA. Further details and hyperparameters are given in Section 4.1.

**Genetic operators.** Key to the success of any GA is the design of its genetic operators; our custom crossover and mutation operators enable SynGA to operate directly over synthesis trees (Figure 1). Given two parent trees $T_1, T_2 \in \mathcal{T}$, we perform crossover by enumerating their subtrees $\mathcal{S}_1, \mathcal{S}_2 \subseteq \mathcal{T}$. Then, we sample a random $(S_1, S_2) \in \mathcal{S}_1 \times \mathcal{S}_2$ that are compatible with at least one bimolecular reaction $R \in \mathcal{R}$ in the sense that $R(M(S_1), M(S_2)) \neq \varnothing$, where $M(T)$ is the final product of $T$. If such a pair exists, we join $(S_1, S_2)$ at a new root node by sampling a random compatible reaction and product. Our crossover is motivated by the intuition that if $S$ is a subtree of $T$, then $M(S)$ is similar to a fragment of $M(T)$. Our crossover corresponds roughly to fusing one fragment from each parent, which has been shown to be effective in synthesis-agnostic GAs (Jensen, 2019; Tripp & Hernández-Lobato, 2023).

Given a tree $T \in \mathcal{T}$, mutation randomly performs one of five operations:

- **Grow.** Apply a random reaction $R \in \mathcal{R}$ compatible with $M(T)$, and choose a random product. If $R$ is bimolecular, then we also sample a building block compatible with $M(T)$ and $R$. The root of $T$ becomes the child of the mutant tree's root.

- **Shrink.** Randomly restrict to one of the subtrees rooted at the children of the root of $T$.

- **Rerun.** Keeping the blocks and reactions fixed, randomly reassign the intermediate products, i.e., $M_v$ for internal nodes $v$. Conceptually, we execute $T$ in a bottom-up (forward) direction but instead of selecting a single product $M_v \in \mathcal{M}_S$ per reaction, we maintain and propagate sets of intermediates $\mathcal{M}_v \subseteq \mathcal{M}_S$, such that $\mathcal{M}_v = \bigcup R_v(\{M_w \in \mathcal{M}_w \mid w \text{ is a child of } v\})$. This yields a set of alternate products $\mathcal{M}_{\text{root}} - \{M(T)\}$ that can be produced using $T$. We randomly pick one of them and backtrack to resolve the intermediates leading to it. In practice, Rerun is implemented using a one-pass algorithm that streams the reassignments in random order, such that intermediates are only ever materialized on demand.

- **Change internal.** Randomly change the reaction assigned to an internal node to a new one that is compatible with its children. Rerun to obtain the mutant tree.

- **Change leaf.** Randomly change the block assigned to a leaf to a new one that is compatible with its parent and sibling (if any). Rerun to obtain the mutant tree.

Grow and Shrink are picked with probability $0.125$ and the others with probability $0.25$. Grow can also be used to sample full synthesis trees by repeatedly applying it to a random building block for a random number of steps. We use this to initialize the population in SynGA and to generate datasets for ML. Appendix A.2 discusses additional implementation details that we omitted for clarity.

**Fitness functions.** A strength of GAs is their flexibility in the choice of fitness function $f$, allowing us to support both property optimization and analog search under a unified framework. For the former, the goal is to maximize a property $\rho$ of interest, so we can simply set $f = \rho$. For analog search, $f$ can be taken to be some notion of similarity to the query molecule. Although both problems can be framed as fitness maximization, a key distinction is that in property optimization, the fitness function is treated as an opaque oracle for which sample-efficiency is a priority. In contrast, in analog search, one can evaluate the fitness function trivially and the task's goal (the query molecule) is known. This difference impacts how we can interface SynGA with ML.

## 3.2 BUILDING BLOCK FILTERING

We propose an elegant ML complement to SynGA that is deep block filtering. In the context of analog search, we learn a network $\pi_\theta \colon M \mapsto \mathcal{F}_M \subseteq \mathcal{B}$ that selects the most relevant building blocks $\mathcal{F}_M$ to some query molecule $M$. If $|\mathcal{F}_M| \ll |\mathcal{B}|$, filtering can be highly effective, especially since our experiments use a catalog of almost 200k blocks. To search for an analog of $M$, SynGA can then be run using $\mathcal{F}_M$ instead of $\mathcal{B}$. However, we consider an $\varepsilon$-filtered approach to account for potential errors made by $\pi_\theta$. When a block is to be sampled from a space $\mathcal{S} \subseteq \mathcal{B}$, we instead sample from the filtered intersection $\mathcal{S} \cap \mathcal{F}_M$ with probability $1 - \varepsilon$ (if nonempty), and the original subset $\mathcal{S}$ otherwise. We set $\varepsilon = 0.1$ in all our experiments.

Since $\mathcal{B}$ is large and discrete, parameterizing a $\pi_\theta$ that selects from it is challenging. Prior work has approached this problem by using a diffusion model that generates fingerprints and then performing

nearest neighbor searches (Gao et al., 2024). Instead, we frame the problem as a classification task. That is, we learn a binary classifier $\pi_\theta : \mathcal{M} \times \mathcal{B} \to (0, 1)$ that predicts whether a block can be used to produce a molecule. Then, we can filter $\mathcal{F}_M = \{\pi_\theta(M, B) > \mu \mid B \in \mathcal{B}\}$, for some threshold $\mu$. Since $B$ is given as input, the model has explicit access to the structure of $\mathcal{B}$ and does not have to learn it implicitly. This allows us to use a much smaller multilayer perceptron (MLP) model while achieving strong performance. We train $\pi_\theta$ on a dataset $\mathcal{D} = \{(M, \mathcal{B}_M)\}$ of product-block(s) pairs obtained by randomly sampling millions of synthesis routes. We use the binary cross entropy loss and resample the positive set $\mathcal{B}_M$ and negative set $\mathcal{B} - \mathcal{B}_M$ with equal probability, since the former is orders of magnitude smaller. Further details on architecture and training are given in Section 4.2.1. Although $\pi_\theta$ is trained on exact product-block pairs, we use $\pi_\theta(M, B)$ to predict whether $B$ can produce an analog of $M$, if $M$ is unsynthesizable.

### 3.3 BLOCK ADDITIVE MODELS

For property optimization, the above approach to block filtering is infeasible since generating a large dataset would be too sample-inefficient. Moreover, the tasks do not have explicit goal states, so the classification formulation is less applicable. Thus, we approach block filtering by fitting a neural additive model (NAM) (Agarwal et al., 2021) over a synthesis route's building blocks. Our choice is motivated by the simplicity and intrinsic interpretability of NAMs. Formally, given a property $\rho$ and product-block(s) pair $(M, \mathcal{B}_M)$, the NAM models $\rho(M)$ using a sum of block-wise scores:

$$\rho_\theta(\mathcal{B}_M) = \left( \alpha + (1 - \alpha)|\mathcal{B}_M|^{-1} \right) \sum_{B \in \mathcal{B}_M} s_\theta(B). \tag{1}$$

Here, $s_\theta : \mathcal{B} \to \mathbb{R}$ is an MLP and $\alpha \in [0, 1]$ is a learnable parameter that interpolates between a sum and mean. NAMs are easily interpretable in that each block $B$ is assigned a score $s_\theta(B)$ such that products formed from higher-scoring blocks will have higher predicted property scores. Assuming the NAM is reasonably accurate, we can then obtain a subset of promising blocks $\mathcal{F}_M$ by filtering out the highest-scoring ones.

Although some popular properties for drug discovery are roughly additive, others are complex and non-linear, and hence difficult to accurately model with NAMs (Levin et al., 2023). To mitigate this, we first note that our NAM only needs to be accurate with respect to the *relative* ranking between product scores. Thus, we train it with a pairwise ranking objective (Burges et al., 2005) instead of a regression loss. Second, we couple the NAM with a more powerful predictor that filters the samples post-hoc from SynGA. The predictor can correct errors made by the NAM, and conversely, the NAM imposes a prior on the building block space that allows for more targeted exploration, playing an analogous role to that of a generative model. We find this is sufficient for obtaining state-of-the-art results in our experiments, though future work could explore NAMs with higher-order terms or attribution methods to improve the filter's expressivity.

### 3.4 SYNGBO

These components are then integrated in a broader Bayesian optimization algorithm, which we call SynGBO (Algorithm 2). At each step, we use SynGA with NAM filtering to maximize an acquisition function under a Gaussian process (GP) surrogate. The most fit candidates from this inner loop are evaluated under the true oracle, and the outer loop continues until the oracle budget is consumed. The NAM and GP are also periodically refitted as new samples are discovered. SynGBO runs SynGA as a subroutine for roughly $100\times$ more iterations than the standard version of SynGA, but this does not incur prohibitive cost due to the lightweight nature of SynGA and its parallelizability. Further details are given in Appendix C.4.

## 4 EXPERIMENTS

### 4.1 SETUP

**Building blocks.** We start with 211,220 molecules from the Enamine Building Blocks catalog (US Stock, Oct. 2023) (Enamine, 2023) processed by Luo et al. (2024). We further discard deuterated compounds and those containing elements other than B, Br, C, Cl, F, H, I, N, O, P, S, Se, Si

Table 1: Ablation of different building block filters for SynGA on the validation set and 100 test molecules sampled from ChEMBL. We compare no filtering (**None**), a similarity heuristic (**Sim**), an MLP, and an MLP trained with hard negative mining to enhance precision (**MLP + Mine**).

| Filter | AUPRC | AUROC | RR | Morgan | Scaffold | Gobbi | Subset |
|---|---|---|---|---|---|---|---|
| None | – | – | 0.00 | 0.459 | 0.526 | 0.400 | 196,907 |
| Sim | 0.217 | 0.970 | 0.06 | 0.625 | 0.634 | 0.515 | 9892 |
| MLP | 0.212 | **0.999** | **0.22** | **0.721** | **0.724** | **0.635** | 117 |
| MLP + Mine | **0.764** | **0.999** | 0.20 | 0.664 | 0.671 | 0.570 | 184 |

Table 2: Average similarity scores between 1k molecules from ChEMBL and their proposed analogs. Results for SynNet and ChemProjector are taken from Luo et al. (2024). SynthesisNet and SynFormer results were reproduced with their default parameters, and we use the non-MCMC version ($\tau$ in their paper) for SynthesisNet due to compute limitations.

| Method | Valid | RR | Morgan | Scaffold | Gobbi | Time |
|---|---|---|---|---|---|---|
| SynNet | 0.850 | 0.054 | 0.427 | 0.417 | 0.268 | – |
| SynthesisNet | **1.000** | 0.070 | 0.543 | 0.530 | 0.452 | – |
| ChemProjector | 0.988 | 0.133 | 0.598 | 0.587 | 0.557 | – |
| SynFormer | 0.998 | 0.190 | 0.668 | 0.667 | **0.635** | **80** m |
| SynGA (Sim) | **1.000** | 0.064 | 0.631 | 0.638 | 0.534 | – |
| SynGA (MLP) | **1.000** | **0.196** | **0.711** | **0.694** | 0.623 | 250 m |

(e.g., organometallics). Then, the remaining molecules are sanitized and standardized using RDKit (Landrum et al., 2006). Lastly, removing duplicates and blocks unsupported by any reaction template leaves our final set of 196,907 building blocks.

**Reactions.** We use the reaction set from Gao et al. (2022b), which comprises 91 uni- or bi-molecular reaction templates compiled from Hartenfeller et al. (2011) and Button et al. (2019).

**Genetic algorithm.** We use a population size of 500, offspring size of 5, crossover rate $r_{cross} = 0.8$, mutation rate $r_{mut} = 0.5$, and elitist selection (Algorithm 1). Parents are sampled with probability proportional to their inverse rank, which is a simple approximation to the quantile-based sampling scheme used by MolGA (Appendix A.3). To focus on small molecules, we cap all synthesis routes to at most 5 steps (internal nodes) and all products to a generous upper-bound weight of 1000 Da.

**Fingerprints.** For ML modeling and similarity calculations, we use *count* Morgan fingerprints of radius 2 by default, due to their greater specificity compared to *binary* fingerprints, which can fail to discriminate between substructure repetitions. The Tanimoto similarity between fingerprints $\mathbf{x}$ and $\mathbf{y}$ in both cases is $||\min(\mathbf{x}, \mathbf{y})||_1 / ||\max(\mathbf{x}, \mathbf{y})||_1$, where min and max are applied elementwise. We use 4096-dim. fingerprints for the analog search fitness function and other similarity computations.

## 4.2 Synthesizable Analog Search

### 4.2.1 Block Filtering

We begin by exploring various building block filtering models for analog search. To do so, we consider the smaller-scale task of generating analogs for 100 random molecules drawn from ChEMBL (Zdrazil et al., 2023). This was originally proposed in Gao et al. (2022b) as a challenging task for assessing their model's ability to generalize to "unreachable" queries. Since sample-efficiency is not a primary concern in analog search, we run our GAs with a large initial population of 5k and a total budget of 10k oracle calls. To directly optimize for the evaluation metrics from Luo et al. (2024), we set the fitness function to $0.9 \cdot \text{Morgan} + 0.1 \cdot \text{Murcko}$ for the ChEMBL tasks, which are defined shortly later. For each query $M$, the most fit individual is taken as the proposed analog $A$ for further evaluation, leading to 100 query-analog pairs.

The first two rows of Table 1 are ML-free approaches. **None** is the base SynGA, and **Sim** selects all building blocks with count fingerprints $\mathbf{b}$ such that $||\min(\mathbf{b}, \mathbf{q})||_1 / ||\mathbf{b}||_1 > 0.5$, where $\mathbf{q}$ is the query

Table 3: Projection of $N$ query molecules designed by generative models on 6 tasks. If $y$ and $y'$ are the scores of the query and analog molecules respectively, then $\Delta = y' - y$. We report the mean and standard deviation across queries, and the methods' runtimes in the header. The Valid column pertains to SynFormer and we omit it for SynGA since it always achieves perfect validity.

| Task | $N$ | Valid | SynFormer (70 m) | | SynGA (MLP) (180 m) | |
|---|---|---|---|---|---|---|
| | | | Sim. | $\Delta$ ($\uparrow$) | Sim. | $\Delta$ ($\uparrow$) |
| ALDH1 | 230 | 1.000 | $0.457 \pm 0.173$ | $0.118 \pm 1.190$ | $\mathbf{0.536} \pm 0.119$ | $\mathbf{0.302} \pm 1.134$ |
| ESR_ant | 203 | 1.000 | $0.553 \pm 0.151$ | $0.002 \pm 0.832$ | $\mathbf{0.644} \pm 0.134$ | $\mathbf{0.231} \pm 0.803$ |
| TP53 | 232 | 1.000 | $0.590 \pm 0.173$ | $0.290 \pm 0.660$ | $\mathbf{0.630} \pm 0.144$ | $\mathbf{0.359} \pm 0.574$ |
| O. MPO | 46 | 1.000 | $0.406 \pm 0.163$ | $\mathbf{-0.157} \pm 0.157$ | $\mathbf{0.468} \pm 0.155$ | $-0.162 \pm 0.152$ |
| P. MPO | 41 | 0.976 | $0.503 \pm 0.158$ | $-0.173 \pm 0.195$ | $\mathbf{0.566} \pm 0.116$ | $\mathbf{-0.156} \pm 0.197$ |
| S. Hop | 35 | 1.000 | $0.523 \pm 0.112$ | $-0.360 \pm 0.166$ | $\mathbf{0.594} \pm 0.068$ | $\mathbf{-0.351} \pm 0.136$ |

fingerprint. Intuitively, we threshold on the fraction of local structures in the building block that are present in the query. For metrics, the reconstruction rate **RR** is the fraction of cases where $M = A$. **Morgan**, **Scaffold**, and **Gobbi** are the average similarity between $M$ and $A$ under different metrics, namely, the Tanimoto similarity between the Morgan *bit* fingerprints of the pair and their Murcko scaffolds, and the dice similarity of their pharmacophore fingerprints (Gobbi & Poppinger, 1998). **Subset** is the average size of the restricted block subset $|\mathcal{F}_M|$. Surprisingly, the simple similarity heuristic improves performance significantly and reduces the building blocks by orders of magnitude.

Inspired by this, we train a small fingerprint-based MLP model for filtering (Appendix B.1). We generate a dataset by randomly sampling synthesis routes until 10M unique products are found, and hold out 10k of them for validation. Across the validation set, the per-example **AUROC** and **AUPRC**, with respect to the $\mathcal{B}$-wise filter scores and the binary labels, are averaged and reported in Table 1. The MLP filter obtains strong performance on the validation set and retrieves better analogs on the test set, using a score cutoff of $0.5$. We further characterize the contributions of the GA and filter in Appendix B.2, and in Appendix B.3, we describe how the model's precision can be increased substantially using hard negative mining (Robinson et al., 2021) (**MLP + Mine**) but with degraded performance on ChEMBL.

### 4.2.2 COMPARISONS AGAINST BASELINES

We benchmark SynGA on the 1k molecule ChEMBL task from Luo et al. (2024) in Table 2. For baselines, we consider SynNet (Gao et al., 2022b), SynthesisNet (Sun et al., 2025b), ChemProjector (Luo et al., 2024), and SynFormer (Gao et al., 2024), which are ML models that decode a query directly into synthesis routes, represented as actions in a Markov decision process or a postfix string. The search space of SynGA is most comparable to that of ChemProjector. SynNet and SynthesisNet use the same 91 reaction templates but an older and smaller catalog of 147k Enamine building blocks. In contrast, SynFormer uses an expanded 115 template set which includes trimolecular reactions.

Despite its smaller search space, SynGA (MLP) achieves competitive performance in both reconstruction and analog search. In particular, some methods produce a small fraction of invalid routes (**Valid**), whereas SynGA will always yield valid routes by design of its genetic operators. However, SynGA is over $3\times$ slower than SynFormer (see **Time**, although our analysis has limitations discussed in Appendix B.4). This is expected since amortized methods benefit from little to no searching during inference in exchange for a relatively larger model and expensive training stage. In contrast, SynGA is more lightweight but relies on a more expensive search during inference. Hence, while SynGA is less efficient, it is easier to adopt out of the box on new building blocks and reaction sets. Furthermore, we are able to directly optimize for arbitrary notions of chemical similarity. We discuss the strengths and weaknesses of both methods further in Appendix B.5, including some extended metrics on in-distribution query sets where we find SynFormer outperforms SynGA.

### 4.2.3 PROJECTING STRUCTURE-BASED AND GOAL-DIRECTED MOLECULAR DESIGNS

Many state-of-the-art generative models are synthesis-agnostic and often propose unsynthesizable molecules (Gao & Coley, 2020). Commonly, SAscore (Ertl & Schuffenhauer, 2009) is used to justify

Table 4: Sum of the top-10 AUC scores over the PMO suite. Results are taken from their respective papers, except MolGA, REINVENT, and SynNet are taken from Kim et al. (2024). We average over 5 seeds, except $f$-RAG and SynthesisNet use only 3 and 1 seeds, respectively. Separate tables for SynthesisNet and SynFlowNet are given since they were assessed only on 13 and 2 PMO tasks, instead of the full 22. Task-wise results are given in Appendix C.5.

| Method | Synthesis | AUC |
|---|---|---|
| $f$-RAG | ✗ | 16.301 |
| GPBO | ✗ | 16.304 |
| Genetic GFN | ✗ | 16.078 |
| MolGA | ✗ | 15.686 |
| REINVENT | ✗ | 15.003 |
| SynNet | ✓ | 12.610 |
| SynGA | ✓ | 13.366 |
| SynGBO | ✓ | 16.426 |

| Method | Synthesis | AUC |
|---|---|---|
| SynthesisNet | ✓ | 7.906 |
| SynGA | ✓ | 7.836 |
| SynGBO | ✓ | 9.332 |

| Method | Synthesis | AUC |
|---|---|---|
| SynFlowNet | ✓ | 1.576 |
| SynGA | ✓ | 1.842 |
| SynGBO | ✓ | 1.905 |

the synthetic accessibility of a model's samples. However, it has limitations as a heuristic and, ideally, we not only want to know if a molecule can be made but also how to make it. A nice use case of synthesis models such as SynGA is then to "project" arbitrary designs back onto synthesis route space. Even if the original molecule is already synthesizable, doing so can uncover alternatives that are cheaper or more experimentally feasible to synthesize. In the following, we perform synthesizable hit expansions around molecules produced by structure-based and goal-directed generative models and quantify how their docking and property scores change as a result.

For structure-based design, we sample ligands binding to three receptors (ALDH1, ESR_ant, TP53) from LIT-PCBA (Tran-Nguyen et al., 2020), and for goal-directed design, molecules that optimize for three property oracles (Osimertinib MPO, Perindopril MPO, scaffold hop) from GuacaMol (Brown et al., 2019). For ligand design, we use the *negative* QuickVina2 (Alhossary et al., 2015) docking score, so that all properties are to be maximized. Following Luo et al. (2024), for each receptor, ligands are produced using Pocket2Mol (Peng et al., 2022), and for each GuacaMol oracle, we filter the molecules produced by Gao & Coley (2020) (using three methods (Jensen, 2019; Yoshikawa et al., 2018; Segler et al., 2018) adapted by GuacaMol) that are unsynthesizable under ASKCOS (Coley et al., 2019). We noticed clusters of highly similar molecules in our query set, particularly in the GuacaMol tasks where high-scoring molecules are by definition those similar to some reference molecule and, hence, to each other. Consequently, we greedily cluster our dataset using a similarity cutoff of 0.7, and retain the highest-scoring representative from each cluster.

For each query, we run SynGA (MLP) as in Section 4.2.1 but optimizing only Morgan *count* similarity. We dock or evaluate the five most fit analogs and retain the one with the best property score. In Table 3, we report the average similarity between the queries and analogs as well as their average difference in property score. We compare against SynFormer, using their default sampling parameters and also dropping queries that fail to decode into valid analogs for metric computation. Overall, SynGA finds better analogs in terms of both similarity and score preservation or improvement.

## 4.3 De novo Synthesis-Aware Property Optimization

### 4.3.1 Block Additive Models

We first ablate the utility of NAMs in the data-limited regime in Appendix C.1. On small datasets, we find NAMs are able to achieve good test correlation and bias products towards higher scores through building block filtering. However, by coupling NAMs with a more accurate GP predictor, we are able to surpass the performance of both components in isolation, motivating SynGBO's design.

### 4.3.2 Comparisons Against Baselines: PMO

We run SynGA on the Practical Molecular Optimization (PMO) benchmark (Gao et al., 2022a), implemented by the Therapeutics Data Commons (TDC) (Huang et al., 2021). On the PMO benchmark, algorithms optimize against a suite of 23 tasks within 10k oracle calls. Their suggested metric is the

Table 5: Vina docking scores of the top-100 diverse modes on the LIT-PCBA dataset, averaged over all receptors (Mean). We show the scores for ALDH1, ESR_ant, and TP53 as examples, with the full receptor-wise results in Appendix C.6. Results for baselines are taken from Seo et al. (2025) and Shen et al. (2025). For 3DSynthFlow, we note that its prior preprint version attained a score of $-12.25 \pm 0.43$ on ALDH1. We report the mean and standard deviation over 4 seeds.

| Method | Calls | ALDH1 | ESR_ant | TP53 | Mean |
|---|---|---|---|---|---|
| SynNet | | $-8.81 \pm 0.21$ | $-8.52 \pm 0.16$ | $-5.34 \pm 0.23$ | $-8.22$ |
| BBAR | | $-10.06 \pm 0.14$ | $-9.92 \pm 0.05$ | $-7.05 \pm 0.09$ | $-9.36$ |
| SynFlowNet | 64000 | $-10.69 \pm 0.09$ | $-10.27 \pm 0.04$ | $-7.90 \pm 0.10$ | $-9.99$ |
| RGFN | | $-9.93 \pm 0.11$ | $-9.72 \pm 0.14$ | $-7.07 \pm 0.06$ | $-9.20$ |
| RxnFlow | | $-11.26 \pm 0.07$ | $-10.77 \pm 0.04$ | $-8.09 \pm 0.06$ | $-10.45$ |
| 3DSynthFlow | | $-11.82 \pm 0.04$ | $-11.23 \pm 0.08$ | $-8.41 \pm 0.17$ | $-10.88$ |
| SynGA | 16000 | $-11.97 \pm 0.04$ | $-11.07 \pm 0.20$ | $-8.23 \pm 0.14$ | $-10.80$ |
| SynGBO | | $\mathbf{-12.36} \pm 0.04$ | $\mathbf{-11.68} \pm 0.21$ | $\mathbf{-8.51} \pm 0.20$ | $\mathbf{-11.11}$ |

area under the curve (AUC) of the top-10 molecules, normalized to $[0, 1]$. However, we remove the Valsartan SMARTS task for reasons discussed in Appendix C.5, among other details. For baselines, we use $f$-RAG (Lee et al., 2024), Genetic GFN (Kim et al., 2024), and GPBO (Tripp et al., 2021; Tripp & Hernández-Lobato, 2024), synthesis-agnostic state-of-the-art algorithms that use MolGA as a component. $f$-RAG and Genetic GFN couple GraphGA with retrieval-augmented generation and GFlowNets, while GPBO uses MolGA to optimize an acquisition function in Bayesian optimization. We also report REINVENT (Olivecrona et al., 2017) and MolGA (Tripp & Hernández-Lobato, 2023), which were top-performing at the time of publication of PMO. For a synthesis-aware baseline, we use SynNet (Gao et al., 2022b) which couples a synthesis projection model with a fingerprint-based GA. Table 4 provides our results along with more recent synthesis models, SynthesisNet (Sun et al., 2025b) and SynFlowNet (Cretu et al., 2025), that were only assessed on subsets of PMO. SynGA is competitive with the synthesis-aware algorithms on PMO, but lags behind the synthesis-agnostic algorithms. This may reflect the more constrained nature of SynGA's search space, which is a specific subset of (predicted) synthesizable molecules. To bridge the gap, we turn SynGA into a model-based algorithm SynGBO, inspired by the significant improvement of GPBO over MolGA (Appendix C.4). SynGBO attains state-of-the-art performance on PMO, being competitive with or outperforming even the top unconstrained algorithms.

### 4.3.3 COMPARISONS AGAINST BASELINES: DOCKING

To test SynGA on 3D objectives, we optimize for the UniDock (Yu et al., 2023) Vina docking scores of LIT-PCBA receptors (Tran-Nguyen et al., 2020), following Seo et al. (2025). To prevent reward hacking, we use the average of the QED (Bickerton et al., 2012) and *normalized* Vina score $(-0.1 \cdot \text{Vina})$ as our fitness function, and set a cap of 50 heavy atoms. We collect the top diverse modes, by filtering samples by QED $> 0.5$ and greedily clustering with a similarity threshold of 0.5, and report their average docking scores in Table 5. As baselines, we consider a variety of synthesis-aware methods: SynNet (Gao et al., 2022b), BBAR (Seo et al., 2023), GFlowNets (SynFlowNet, RxnFlow, RGFN) (Cretu et al., 2025; Seo et al., 2025; Koziarski et al., 2024), and 3DSynthFlow (Shen et al., 2025). Surprisingly, SynGA attains better docking scores than all baselines except 3DSynthFlow with only a quarter of the oracle calls. This highlights the sample-efficiency and effectiveness of GAs. We note that 3DSynthFlow jointly designs the synthesis route and binding pose and similarly incorporating 3D information into SynGA (e.g., as in Fu et al. (2022)) could be promising for future work. However, we proceed by augmenting SynGA into the model-based version SynGBO, in line with our approach for PMO, to attain the best docking scores overall. Further details are given in Appendix C.6.

## 5 CONCLUSION

We propose SynGA, a simple synthesis-constrained GA that operates directly on synthesis routes. Within a unified framework of fitness maximization, we demonstrate the effectiveness of our method

at synthesizable analog search and property optimization. SynGA is further enhanced by ML through a lightweight building block filter, which manifests as a classifier trained on millions of synthesis routes for analog search and an interpretable block-additive model for sample-efficient property optimization. The latter leads to the model-based variant SynGBO, which achieves state-of-the-art performance on both the PMO benchmark and docking tasks. However, we note that this is just one possibility, and we expect that SynGA can be readily hybridized with ML in many other ways. We provide an extended outlook in Appendix D.

## REPRODUCIBILITY STATEMENT

Hardware specifications are provided in Appendix E along with our source code, which provides documentation for preparing data and running experiments.

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

# A  GENETIC ALGORITHM

## A.1  PSEUDOCODE

In the following, the product of a synthesis tree $T \in \mathcal{T}$ is denoted $M(T) \in \mathcal{M}_S$. SynGA uses elitist selection where only the fittest among the parents and offspring are retained, and is run until a budget of fitness evaluations is fully consumed.

---

**Algorithm 1** Pseudocode for SynGA.

---

**Require:** Initial size $n_0$, population size $n$, offspring size $m$, crossover rate $r_{\text{cross}}$, mutation rate $r_{\text{mut}}$,
  budget $B$, a fitness function $f \colon \mathcal{M} \to \mathbb{R}$.
1: Sample $\mathcal{P} \subseteq \mathcal{T}$ of size $n_0$
2: $\mathcal{H} \leftarrow \{M(T) \to f(M(T)) \mid T \in \mathcal{P}\}$       ▷ keep track of unique fitness evaluations
3: **while** $|\mathcal{H}| < B$ **do**
4:   $\mathcal{O} \leftarrow \varnothing$
5:  **for** $m$ repeats **do**
6:    Sample $T_1, T_2 \in \mathcal{P}$ without replacement
7:    **if** `rand()` $< r_{\text{cross}}$ **then**
8:      $T \leftarrow$ `crossover`$(T_1, T_2)$
9:      **if** `rand()` $< r_{\text{mut}}$ **then**
10:       $T \leftarrow$ `mutate`$(T)$
11:    **else**
12:      $T \leftarrow$ `mutate`$(T_1)$
13:    **if** $T \neq$ `None` and $M(T) \notin \mathcal{H}$ and $|\mathcal{H}| < B$ **then**
14:      $\mathcal{H}[M(T)] \leftarrow f(M(T))$
15:      $\mathcal{O} \leftarrow \mathcal{O} \cup \{T\}$
16:  $\mathcal{P} \leftarrow$ the $n$ fittest individuals from $\mathcal{P} \cup \mathcal{O}$
17: **return** $\mathcal{H}$

---

## A.2  IMPLEMENTATION DETAILS

**SMARTS.** We leverage the fact that the reaction templates are implemented as SMARTS strings to improve the efficiency of SynGA. At a high level, a SMARTS string matches pattern(s) in the input molecules and defines a transformation over them to yield the product(s). The reaction will proceed with one or more products if and only if the input reactants contain the specified patterns. Concretely, a bimolecular SMARTS reaction is of the syntax `S1.S2>>P`, where `S1` and `S2` encode molecular substructures. The reaction will proceed if and only if one of the input molecules contains `S1` and the other `S2`. Thus, we precompute the necessary substructure matches on the base building blocks and cache them for products during runtime. In doing so, we can efficiently infer whether two molecules can react and which blocks are compatible with a given reaction (or vice versa). This strategy, however, is unlikely to scale to large libraries, although most template-based approaches use relatively small ($\sim$100) template sets to our knowledge.

**Edge cases.** There are a number of cases in which crossover and mutation may fail. For example, a Grow operation may produce a molecule over 1000 Da, or crossover may fail to find two internal nodes that can be linked by a reaction. Depending on the edge case, we employ one of two strategies to handle it: (1) imposing boundary conditions to prevent invalidating operations from being taken, and (2) retrying the same (random) operation up to 10 times, as in GraphGA (Jensen, 2019).

**Parallelization.** Fortunately, GAs are highly amenable to parallelization. In particular, sampling the initial population, crossover and mutation for parent pairs, and evaluating the fitness function over offspring can all be implemented in a parallel manner. Benchmarks that require running multiple GA trials can also be parallelized. We leverage this in our implementation.

## A.3  INVERSE-RANK SAMPLING

MolGA (Tripp & Hernández-Lobato, 2023) samples its mating pool through independent repetitions of the following: first sample $u \sim \mathcal{U}[-3, 0]$ and then sample uniformly from the top $\varepsilon = 10^u$

Table 6: Sum of the top-10 AUC scores for JNK3 and Osimertinib MPO. The last row corresponds to our current hyperparameters. We report the mean and standard deviation over 5 seeds.

| Grow | Shrink | Rerun | CI | CL | AUC |
|------|--------|-------|----|----|-----|
| 0 | 1 | 1 | 1 | 1 | $1.465 \pm 0.068$ |
| 1 | 0 | 1 | 1 | 1 | $1.465 \pm 0.076$ |
| 1 | 1 | 0 | 1 | 1 | $1.496 \pm 0.052$ |
| 1 | 1 | 1 | 0 | 1 | $1.470 \pm 0.067$ |
| 1 | 1 | 1 | 1 | 0 | $1.451 \pm 0.104$ |
| 1 | 1 | 1 | 1 | 1 | $1.457 \pm 0.044$ |
| 1 | 1 | 2 | 2 | 2 | $1.504 \pm 0.131$ |

fraction of the population. To accommodate different population sizes $n$, suppose we instead sample $u \sim \mathcal{U}[-\log_{10}(n), 0]$. Then a change of variables shows that $\varepsilon$ has the density

$$p(\varepsilon) = \frac{1}{\log_{10}(n)} \cdot \left| \frac{d}{d\varepsilon} \log_{10}(\varepsilon) \right| = \frac{1}{Z\varepsilon},$$

supported on $[\frac{1}{n}, 1]$, for some normalization constant $Z$. Now for $1 \leq k \leq n$, the probability $p_k$ of sampling the $k$-th most fit individual is

$$p_k = \int_{1/n}^{1} p(k \,|\, \varepsilon) p(\varepsilon) \, d\varepsilon, \quad \text{where } p(k \,|\, \varepsilon) = \begin{cases} 1/\lfloor n\varepsilon \rfloor, & \text{if } \varepsilon \geq k/n, \\ 0, & \text{otherwise.} \end{cases}$$

Since $n$ is large and $1/\lfloor n\varepsilon \rfloor \approx 1/n\varepsilon$, we can approximate that

$$p_k \approx \int_{k/n}^{1} \frac{1}{Zn\varepsilon^2} \, d\varepsilon \propto \frac{1}{k} - \frac{1}{n} \approx \frac{1}{k}.$$

Hence, the sampling strategy used in MolGA can be well-approximated by sampling proportionally to each individual's inverse rank. We use inverse-rank sampling for SynGA due to its simplicity, especially when sampling without replacement. Lastly, we note that the more general distribution

$$p_k \propto \frac{1}{k + \lambda n},$$

has been proposed in prior work (Kim et al., 2024), although its mathematical connection to MolGA was not made explicit. Here, setting $\lambda = 0$ recovers inverse-rank sampling.

### A.4 MUTATION ABLATIONS

Mutation probabilities are set proportionally to an assigned weight for each of the five actions. In our experiments, we assign a weight of 1 to Grow and Shrink, but 2 to Rerun, Change Internal (CI), and Change Leaf (CL) operation probabilities (i.e., making them twice as likely), since we expected the latter to produce more local perturbations. As a sensitivity ablation, we explore various action weights on two tasks from the PMO benchmark, following the setup in Section 4.3.2. Table 6 shows that SynGA is robust across multiple settings. We acknowledge that our ablation may not necessarily extend to other property functions, though hyperparameter tuning on the full PMO suite would also likely be overfitting.

## B SYNTHESIZABLE ANALOG SEARCH

### B.1 CLASSIFIER MODELLING

We use a five-layer MLP of width 256 that takes as input $[\mathbf{q}, \mathbf{b}, \min(\mathbf{q}, \mathbf{b})] \in \mathbb{N}_0^{3d}$, where $d = 2048$ and $\mathbf{q}$ and $\mathbf{b}$ are the Morgan count fingerprints of the query and building block, respectively. We use GELU activations and batch normalization. The MLP has 1.8M parameters but 1.6M of them are allocated to the first layer. The MLP is trained for 500k steps using the Adam (Kingma & Ba, 2015) optimizer with learning rate $5 \times 10^{-4}$, and batch size 1024. We opt for an MLP for its simplicity and efficiency; the success of the fingerprint-similarity heuristic suggested that such a network could work well in the first place. We leave exploring more sophisticated architectures for future work.

Table 7: Ablation of SynGA versus pure random search on base and MLP-filtered building blocks.

| Method | Filter | RR | Morgan | Scaffold | Gobbi |
|--------|--------|------|--------|----------|-------|
| Random | None | 0.00 | 0.277 | 0.364 | 0.288 |
|        | MLP  | 0.14 | 0.619 | 0.629 | 0.534 |
| SynGA  | None | 0.00 | 0.459 | 0.526 | 0.400 |
|        | MLP  | 0.22 | 0.721 | 0.724 | 0.635 |

Table 8: Extended comparison of SynGA against SynFormer on ChEMBL and their proposed analogs. We run SynGA for 3.3k iterations to roughly tie the runtime.

| Method | Valid | RR | Morgan | Scaffold | Gobbi | Time |
|--------|-------|------|--------|----------|-------|------|
| SynFormer | 0.998 | 0.190 | 0.668 | 0.667 | **0.635** | 80m |
| SynGA (MLP, 10k) | **1.000** | **0.196** | **0.711** | **0.694** | 0.623 | 250m |
| SynGA (MLP, 3.3k) | **1.000** | 0.175 | 0.690 | 0.675 | 0.609 | 70m |

## B.2 RANDOM SEARCH OVER FILTERED BUILDING BLOCKS

We quantify the contribution of SynGA on the 100 molecule ChEMBL task in Table 7. Instead of running SynGA, we sample an additional 5k synthesis routes (**Random**) under both the base and MLP-filtered building blocks. Our results suggest that, while filtering alone works reasonably well for analog search, it couples synergistically with SynGA to produce even stronger performance.

## B.3 HARD-NEGATIVE MINING

To improve the MLP filter's precision, we explore hard-negative mining from the contrastive learning literature (Robinson et al., 2021). Given a molecule $M$ and building blocks $\mathcal{B}_M \subseteq \mathcal{B}$ that can produce it, we draw negative samples uniformly from $\mathcal{B} - \mathcal{B}_M$. Since $|\mathcal{B}_M| \ll |\mathcal{B}|$, its complement includes many blocks that are highly dissimilar to those in $\mathcal{B}_M$, i.e., "easy" negatives. Thus, we obtain more targeted negative examples by precomputing the 100 most similar blocks $\mathcal{N}(B)$ to each block $B$. Then, after selecting a positive example $B_1 \in \mathcal{B}_M$, we sample the negative example from $\mathcal{N}(B_1) - \mathcal{B}_M$ with probability 0.5, and $\mathcal{B} - \mathcal{B}_M$ otherwise. Negative mining (**MLP + Mine** in Table 1) significantly improves the model's precision on the validation set, but decreases performance on ChEMBL. We attribute this to two potential reasons: (1) test molecules in ChEMBL may be out-of-distribution since the model is only ever shown "reachable" examples, and (2) our negative set may contain false negatives, as we can never know with certainty that a given block *cannot* produce a given molecule (only that certain blocks do). This underscores the limitations behind using performance on our classification task as a direct indicator of filter quality.

## B.4 RUNTIME BENCHMARKING

Experiments were run on an NVIDIA RTX A6000 GPU and a 64-core AMD Ryzen Threadripper PRO 3995WX processor. For SynFormer, we use 12 workers since too many resulted in the GPU going out of memory. For SynGA, we use 100 workers parallelized across the batch dimension, so that each query is run with 1 worker. In general, we expect SynFormer to benefit more from better GPU compute since it requires multiple inference calls with a large ML model, in contrast to SynGA which is predominantly CPU-bound.

**Limitations.** Our runtime metrics in Table 2 should only be taken as a rough estimate. For future work, a more careful analysis could explore different environments and inputs over multiple trials. Also of note is that SynGA and SynFormer are research projects, whose codebases are not written with maximal efficiency in mind. It is likely that the efficiency of both methods can be improved through better engineering.

Table 9: Extended comparisons of SynGA against baselines on three datasets. ChemProjector results on DDS-10 and ZINC were obtained using the same parameters as SynFormer. We also report the average diversity within the top 5 analogs for each query.

| Dataset | Method | Valid | RR | Morgan | Scaffold | Gobbi | Diversity |
|---------|--------|-------|-----|--------|----------|-------|-----------|
| DDS-10 | ChemProjector | 0.988 | 0.657 | 0.882 | 0.881 | 0.853 | 0.272 |
| | SynFormer | **1.000** | **0.858** | **0.963** | **0.966** | **0.949** | 0.243 |
| | SynGA (MLP) | **1.000** | 0.707 | 0.930 | 0.924 | 0.869 | 0.272 |
| ZINC | ChemProjector | 0.992 | 0.379 | 0.744 | 0.757 | 0.686 | 0.323 |
| | SynFormer | 0.999 | **0.558** | **0.841** | **0.845** | **0.806** | 0.290 |
| | SynGA (MLP) | **1.000** | 0.459 | 0.828 | 0.826 | 0.723 | 0.280 |
| ChEMBL | ChemProjector | 0.988 | 0.133 | 0.598 | 0.587 | 0.557 | – |
| | SynFormer | 0.998 | 0.190 | 0.668 | 0.667 | **0.635** | 0.343 |
| | SynGA (MLP) | **1.000** | **0.196** | **0.711** | **0.694** | 0.623 | 0.246 |

## B.5 EXTENDED COMPARISONS AGAINST BASELINES

**Budget Tying.** In Table 8, we compare SynGA against SynFormer but attempt to tie the wall clock time. To do so, we run SynGA for 3.3k oracle calls to roughly match the original 3× slowdown of SynGA to SynFormer. The initial population size is also scaled down proportionally and all other hyperparameters are unchanged. We find that even with a third of the budget, SynGA exhibits strong (though slightly degraded) performance on analog search.

**Additional Test Sets.** In Table 9, we compare SynGA against the top two baselines from Table 2 on two additional datasets: 1k molecules randomly sampled from the Enamine Discovery Diversity Set (DDS-10) (Enamine, 2025) and ZINC (Irwin et al., 2020). SynGA consistently outperforms ChemProjector, but SynFormer outperforms SynGA on DDS-10 and ZINC. For the similarity metrics (Morgan and Scaffold) that we optimize for, the improvement is relatively modest ($\sim$0.03), but the reconstruction rate of SynFormer is considerably higher ($>$10%). SynFormer has been extensively trained for molecular reconstruction (i.e., retrosynthesis), whereas SynGA has only been trained on a block relevance task. Finding the precise synthesis route that produces a molecule is a needle-in-a-haystack task that may be challenging for GAs that directly search over synthesis space (we filter the top 1000 blocks, but this is still a very large search space). In contrast, "projection" is a smoother optimization landscape that can be more effectively navigated by GAs.

The high reconstruction and similarity scores on DDS-10 and ZINC also suggests that both datasets are highly similar to the search spaces of all methods, in contrast to datasets like ChEMBL and the ML-designed molecules from Section 4.2.3. DDS-10 is particularly "reachable", with both SynGA and SynFormer achieving $>$0.9 average similarity, which aligns with DDS-10 and both methods' building blocks coming from Enamine (SynFormer also incorporates Enamine reaction templates). We hypothesize that out-of-distribution datasets pose problems to amortized methods similar to our findings with negative mining in Appendix B.3, which SynGA can overcome through its extensive search process. Given the prior discussion, a promising direction is then to couple SynGA with ML projection or retrosynthesis models to make up for the weaknesses of both methods. We propose some concrete ideas of this in our extended outlook (Appendix D).

## C DE NOVO SYNTHESIS-AWARE PROPERTY OPTIMIZATION

### C.1 NAM ABLATIONS

We first explore the utility of NAMs in the data-limited regime on two property oracles, JNK3 (Li et al., 2018) and Osimertinib MPO (Brown et al., 2019). For each, we run SynGA for 1100 oracle calls and hold out the last 100 discovered molecules as our test set. We apply a random 9:1 training-validation split of the first 1000 molecules, fit a NAM on the training set (Appendix C.2), and filter the top 1000 highest-scoring building blocks under the NAM. Then, we sample 100 synthesis routes and measure their average property score (**Score**). We also measure the Spearman correlation

Table 10: Test correlation and sample score of additive models on two oracles. Coupling the NAM with a more accurate predictor (GP) improves the score of sampled molecules. The mean and standard deviation over 5 seeds is reported.

| | JNK3 | | Osimertinib MPO | |
| Model | Corr. | Score | Corr. | Score |
|---|---|---|---|---|
| Random | − | $0.058 \pm 0.002$ | − | $0.132 \pm 0.023$ |
| Oracle (Sum) | $0.739 \pm 0.127$ | $0.185 \pm 0.010$ | $0.408 \pm 0.159$ | $0.497 \pm 0.021$ |
| Oracle (Mean) | $0.411 \pm 0.183$ | $0.185 \pm 0.010$ | $0.444 \pm 0.106$ | $0.497 \pm 0.021$ |
| NAM | $0.877 \pm 0.043$ | $0.115 \pm 0.016$ | $0.647 \pm 0.070$ | $0.603 \pm 0.027$ |
| GP | $\mathbf{0.959} \pm 0.017$ | $0.188 \pm 0.009$ | $\mathbf{0.847} \pm 0.055$ | $0.679 \pm 0.005$ |
| NAM + GP | − | $\mathbf{0.297} \pm 0.050$ | − | $\mathbf{0.714} \pm 0.020$ |

between the predicted and true property scores over the test set; we choose this metric because our block filtering and SynGA are rank-based methods (e.g., elitist selection depends only on rank).

As seen in the first section of Table 10, the NAM is able to achieve good test correlation and sampling from its filtered building blocks biases products towards higher scores, compared to random sampling from an unfiltered block set (**Random**). We also report **Oracle**, which takes $s_\theta$ in Equation 1 to be the property function and $\alpha = 1$ (Sum) or $\alpha = 0$ (Mean). This can be thought of an idealized model that measures how additive the property functions are. The NAM's performance can be pushed further by coupling it with a stronger predictive model. We fit a Gaussian process (GP) to the training set (Appendix C.3), sample 10k synthesis routes, and use the posterior mean to select 100 products for further evaluation. Their average scores are given in the second section along with the GP's test set correlation. **NAM + GP** samples from the NAM-filtered building blocks, whereas **GP** uses just the base blocks. As expected, the GP is more accurate than the NAM, but by coupling the two we are able to surpass the performance of both components in isolation.

## C.2 NAM MODELLING

To implement the NAM $s_\theta$, we use a five-layer MLP of width 64 that takes as input the 2048-count Morgan fingerprint of the input block. We use GELU activations and no normalization. The NAM has 140k parameters. We train the NAM using the Adam (Kingma & Ba, 2015) optimizer with learning rate $5 \times 10^{-4}$, batch size 50, and early stopping on the validation Spearman correlation with a 5 epoch patience. We use the RankNet (Burges et al., 2005) objective which computes the loss as:

$$\mathcal{L}(\theta) = \text{BCE}\Big( \rho_\theta(\mathcal{B}_1) - \rho_\theta(\mathcal{B}_2), \ \mathbb{I}\left[\rho(M_1) > \rho(M_2)\right] \Big),$$

for a pair of examples $(M_1, \mathcal{B}_1)$ and $(M_2, \mathcal{B}_2)$, where $\rho_\theta$ is the NAM and $\rho$ is the property function and $\mathbb{I}$ is the indicator function. We average the loss over pairwise combinations of the batch.

We found ranking loss to outperform the standard mean-squared-error (MSE) loss for both the NAM and NAM + GP models (Table 11). The ranking objective leads to better NAM test correlation on both objectives. On Osimertinib MPO, this translates to an increase in sample scores. On JNK3, scores are marginally worse, which we hypothesize is because the JNK3 oracle is already well-approximated by additive models, as shown in Table 10.

## C.3 GAUSSIAN PROCESSES

Our Gaussian process uses 2048-count Morgan fingerprints as the features, the MinMax kernel from Gauche (Griffiths et al., 2024), and GPytorch (Gardner et al., 2018) for its implementation.

## C.4 SYNGBO

Inspired by GPBO (Tripp & Hernández-Lobato, 2024), we convert SynGA into a model-based variant SynGBO. At a high level, SynGBO uses SynGA to optimize an acquisition function within a broader Bayesian optimization loop. At each step, a GP and NAM are fit to the samples discovered thus far,

Table 11: Ablation of MSE versus ranking loss for NAM training. The mean and standard deviation over 5 seeds is reported.

| Model | Loss | JNK3 | | Osimertinib MPO | |
|---|---|---|---|---|---|
| | | Corr. | Score | Corr. | Score |
| NAM | MSE | $0.864 \pm 0.035$ | $0.118 \pm 0.023$ | $0.458 \pm 0.110$ | $0.450 \pm 0.058$ |
| | Rank | $\mathbf{0.877} \pm 0.043$ | $0.115 \pm 0.016$ | $\mathbf{0.647} \pm 0.070$ | $0.603 \pm 0.027$ |
| NAM + GP | MSE | – | $\mathbf{0.301} \pm 0.059$ | – | $0.709 \pm 0.007$ |
| | Rank | – | $0.297 \pm 0.050$ | – | $\mathbf{0.714} \pm 0.020$ |

following Appendix C.2 and C.3. Since GPs scale poorly with dataset size, we subset to the top 2500 samples and a random subset of 2500 other samples in practice, and to avoid repeated retrainings, we only refit the NAM every 25 steps. Then, we use SynGA to optimize an acquisition function under the GP surrogate. Following Tripp & Hernández-Lobato (2024), we use the upper confidence bound (UCB) acquisition with $\beta \sim [0.01, 1]$ sampled logarithmically until 5000 samples are obtained, after which we set $\beta = 0$ and maximize the posterior mean. The process repeats until a budget of oracle calls is exhausted (Algorithm 2).

In the inner loop, we run SynGA for 5 generations with an offspring size of 100. We use a population size of 1000 starting with 500 randomly sampled individuals and the top 1000 scoring molecules. All other parameters are kept the same as Section 4.1. In total, SynGA proposes $500 + 5 \cdot 100 = 1000$ *new* molecules at most, of which the 10 most fit ones are evaluated by the true oracle. Hence, SynGA proposes 100 molecules for every molecule evaluated. In contrast, GPBO proposes roughly 1000. As noted by Tripp & Hernández-Lobato (2024), performance can likely be improved by increasing the number of outer and inner loop iterations of SynGBO.

---

**Algorithm 2** Pseudocode for SynGBO.

---

**Require:** Proposal size $m$, budget $B$, GP $g$, NAM $s_\theta$, a fitness function $f \colon \mathcal{M} \to \mathbb{R}$.
1: $\mathcal{H} \leftarrow \{M(T_i) \to f(M(T_i)) \mid T_i \in \mathcal{T}\}$ for $m$ samples
2: $i \leftarrow 0$
3: **while** $|\mathcal{H}| < B$ **do**
4:     **if** $|\mathcal{H}| \geq 500$ and $i \equiv 0 \pmod{25}$ **then**
5:         Fit $s_\theta$ to $\mathcal{H}$
6:         $\mathcal{F} \leftarrow$ top-1000 scoring blocks in $\mathcal{B}$ under $s_\theta$
7:     **else**
8:         $\mathcal{F} \leftarrow \mathcal{B}$
9:     Fit $g$ to $\mathcal{H}$, or a subset if $|\mathcal{H}|$ is large
10:     $\mathcal{P}_0 \leftarrow$ top-1000 candidates in $\mathcal{H}$ and 500 random routes
11:     $\alpha \leftarrow \mathrm{UCB}(g, \beta)$ for $\beta \sim p(\beta)$
12:     Run SynGA with filtered blocks $\mathcal{F}$ from initial population $\mathcal{P}_0$ with fitness function $\alpha$
13:     **for** the $m$ fittest individuals $T$ from SynGA **do**
14:         $\mathcal{H}[M(T)] \leftarrow f(M(T))$
15:     $i \leftarrow i + 1$
16: **return** $\mathcal{H}$

---

### C.5 PRACTICAL MOLECULAR OPTIMIZATION BENCHMARK

The top-$k$ AUC PMO metric is formally defined as $\frac{1}{B} \sum_{t=1}^{B} \bar{\rho}_{k,t}$, where $B$ is the budget and $\bar{\rho}_{k,t}$ is the average of the top $k$ oracle scores within the first $t$ samples. In Gao et al. (2022a), this is further estimated using the trapezoidal rule at 100 sample intervals. Tables 14, 15, and 16 give the expanded task-wise results of Table 4.

**PyTDC.** The original PMO implementation from Gao et al. (2022a) used PyTDC 0.3.6 (Huang et al., 2021). On 0.3.7 onwards, PyTDC made a bug fix[1] that led to breaking changes in the Isomers,

---

[1]https://github.com/mims-harvard/TDC/pull/171

Table 12: Number of discovered modes (i.e., docking score < −10, QED > 0.5, similarity threshold of 0.5) for the ALDH1 task. Results for baselines are taken from Shen et al. (2025). We report the mean and standard deviation over 4 seeds.

| Method | 1k Calls | 5k Calls | 10k Calls |
|--------|----------|----------|-----------|
| RxnFlow | $4.5 \pm 2.1$ | $26.5 \pm 7.8$ | $73.5 \pm 33.2$ |
| 3DSynthFlow | $18.5 \pm 14.8$ | $112.0 \pm 94.8$ | $326.5 \pm 316.1$ |
| SynGA | $32.5 \pm 1.5$ | $144.2 \pm 16.2$ | $241.5 \pm 29.0$ |
| SynGBO | $50.5 \pm 18.5$ | $171.8 \pm 36.8$ | $182.0 \pm 30.9$ |

Sitagliptin MPO, and Zaleplon MPO oracles. For example, Kim et al. (2024) reproduce PMO with PyTDC 0.4.0, and we observe a consistent increase in PMO scores in their results. Thus, care must be taken to pin the PyTDC version when using numbers from Gao et al. (2022a). However, we found PyTDC 0.3.6 difficult to install due to dependency conflicts and its pins to overly old versions of some libraries. In the interest of using up-to-date packages, we use PyTDC 1.1.14 and avoid numbers from the original PMO paper. We also spot check 10k molecules from ZINC (Irwin et al., 2020) to confirm that there are no discrepancies between PyTDC 1.1.14 and 0.3.6 for oracles other than those mentioned above.

**Valsartan SMARTS.** Many methods fail to optimize the Valsartan SMARTS task. This is because the oracle returns 0 if the input does not contain `CN(C=O)Cc1ccc(c2ccccc2)cc1`. That is, models are given no signal (i.e., the oracle appears constant) until they propose a molecule with one specific substructure. But a priori, without any signal or information about the task, there is no reason an algorithm should do so. For this reason, we argue Valsartan SMARTS is ill-suited for benchmarking and remove it.

**Extended Results.** Table 17 reports other metrics for SynGA and SynGBO, for completeness, and Figure 2 provides some optimization curves.

## C.6 LIT-PCBA DOCKING BENCHMARK

To better leverage the GPU batching of UniDock, we increase the offspring size of SynGA to 100 and the proposal size of SynGBO to 20. Since the benchmark emphasizes diversity, we further increase the population size of SynGA to 5000 and initial population size to 1000. Table 18 gives the expanded receptor-wise results of Table 5.

**Extended Results.** Table 19 reports the average ligand efficiency (Vina score normalized by heavy atom count) of the top modes. For a diversity-focused metric, we report the number of discovered ALDH1 modes as a function of oracle calls in Table 12. SynGA and SynGBO perform more diverse exploration than baselines in the early stages of optimization but fall behind 3DSynthFlow afterwards. We attribute this to the exploitative nature of GAs whose goal is property optimization, whereas GFlowNets and CGFlow instead aim to sample proportionally to the property or reward. Example ligands proposed by SynGA and SynGBO are displayed in Figures 3 and 4, which we obtain from the top mode across all seeds that passes the Tartarus (Nigam et al., 2023) filters and whose ring systems all appear in ChEMBL at least 5 times.[2] For future work, many of these filters can be applied on the block- or GA-level to minimize the number of post-hoc rejected samples.

Finally, we perform a preliminary ablation aiming to match the search space of 3DSynthFlow. We consider both the 1.2M and 300k block sets mentioned in their work, which are obtained from Enamine's comprehensive and in-stock catalogs, respectively. For the former, only the current-day catalog version is available and, since it has grown to 1.6M blocks, we randomly downsample it to 1.2M blocks. In both cases, we use the same 38 bimolecular reaction templates, 2 maximum reaction steps, and 40 maximum atoms. We report our results on two receptors in Table 13. All hyperparameters are kept the same, except we filter the top 10k blocks when using the 1.2M catalog set to accommodate for the larger block space. We find the general trend holds: using only a quarter of the oracle calls, SynGBO outperforms or competes with 3DSynthFlow, and SynGA outperforms or competes with all other baselines.

---

[2]https://github.com/PatWalters/useful_rdkit_utils

Table 13: Vina docking scores of the top-100 diverse modes on two LIT-PCBA receptors, over various search spaces. Only the top 3 baselines are shown for conciseness. We report the mean and standard deviation over 4 seeds.

| Method | Search Space | ESR_ant | TP53 |
|---|---|---|---|
| SynFlowNet | | $-10.27 \pm 0.04$ | $-7.90 \pm 0.10$ |
| RxnFlow | | $-10.77 \pm 0.04$ | $-8.09 \pm 0.06$ |
| 3DSynthFlow | | $-11.23 \pm 0.08$ | $-8.41 \pm 0.17$ |
| SynGA | Ours | $-11.07 \pm 0.20$ | $-8.23 \pm 0.14$ |
| SynGA | 3DSynthFlow (1.2M) | $-10.92 \pm 0.15$ | $-8.17 \pm 0.25$ |
| SynGA | 3DSynthFlow (300k) | $-10.89 \pm 0.13$ | $-8.24 \pm 0.13$ |
| SynGBO | Ours | $-11.68 \pm 0.21$ | $-8.51 \pm 0.20$ |
| SynGBO | 3DSynthFlow (1.2M) | $-11.39 \pm 0.12$ | $-8.72 \pm 0.33$ |
| SynGBO | 3DSynthFlow (300k) | $-11.53 \pm 0.04$ | $-8.45 \pm 0.41$ |

## D    EXTENDED OUTLOOK

**Limitations.** SynGA inherits the general limitations of template-based approaches to synthesis. For example, our templates do not *guarantee* synthesizability, nor do they consider reaction conditions, stereochemistry, yield, or cost. Any fixed template library also necessarily restricts exploration to a biased subset of synthesizable space. This can be mitigated by enlarging the template set, though a naïve extension may degrade efficiency and robustness. In addition, the tasks considered in this work are single-objective (or scalarized) whereas real-world molecular design is highly multi-objective. Fortunately, multi-objective GAs such as NSGA-II (Deb et al., 2002) can be easily integrated with our proposed genetic operators. Many of the tasks are also synthesis-agnostic and we treat synthesizability as an additional dimension that contextualizes our results. Evaluating SynGA on benchmarks such as Tartarus (Nigam et al., 2023) that attempt to penalize "unreasonable" samples could enrich our comparisons with synthesis-agnostic baselines for future work.

**Future directions.** While we present SynGA as a standalone work, our hope is that SynGA can also serve as a building block for future ML algorithms. For analog search, one prospective direction could be to refine the outputs of synthesis models like SynFormer (Gao et al., 2024) by running SynGA briefly, and to potentially even improve the model by finetuning on the better analogs. For property optimization, SynGA can be used to boost exploitation or exploration in generative models. Augmenting SynGA with ML may also be promising, such as using a 3D network to enhance the genetic operators for docking tasks. Finally, we note that SynGA is just one synthesis-constrained GA, and future work can look into exploring the rich design space of genetic operators.

## E    REPRODUCIBILITY

**Compute.** All experiments were run on a single NVIDIA RTX A6000 GPU and a 64-core AMD Ryzen Threadripper PRO 3995WX processor. The compute used for analog search is discussed in Appendix B.4 and runtimes are given in Table 2. Training the MLP block filter took ∼4 hours. For PMO, SynGA took ∼20 min per trial with 5 workers and SynGBO took ∼6 hours per trial with 20 workers. However, we ran many trials concurrently on the same machine, so these times are likely inflated. For the docking experiments, SynGA took ∼3 hours with 50 workers and SynGBO took ∼4 hours with 20 workers. We ran trials sequentially and we found that computing the docking scores was a significant portion of the runtime.

**Code.** https://github.com/alstonlo/synga.

Table 14: Task-wise results of Table 4.

| Oracle Synthesis | f-RAG ✗ | GPBO ✗ | G. GFN ✗ | SynGA ✓ | SynGBO ✓ |
|---|---|---|---|---|---|
| Albu. Sim. | 0.977 ± 0.002 | 0.964 ± 0.050 | 0.949 ± 0.010 | 0.649 ± 0.058 | 0.947 ± 0.024 |
| Amlo. MPO | 0.749 ± 0.019 | 0.720 ± 0.061 | 0.761 ± 0.019 | 0.573 ± 0.019 | 0.670 ± 0.088 |
| Cele. Redisc. | 0.778 ± 0.007 | 0.860 ± 0.002 | 0.802 ± 0.029 | 0.494 ± 0.063 | 0.856 ± 0.013 |
| Deco Hop | 0.936 ± 0.011 | 0.672 ± 0.118 | 0.733 ± 0.109 | 0.629 ± 0.014 | 0.831 ± 0.039 |
| DRD2 | 0.992 ± 0.000 | 0.902 ± 0.117 | 0.974 ± 0.006 | 0.976 ± 0.006 | 0.981 ± 0.010 |
| Fexo. MPO | 0.856 ± 0.016 | 0.806 ± 0.006 | 0.856 ± 0.039 | 0.773 ± 0.018 | 0.833 ± 0.018 |
| GSK3β | 0.969 ± 0.003 | 0.877 ± 0.055 | 0.881 ± 0.042 | 0.866 ± 0.072 | 0.924 ± 0.027 |
| Isom. C7H8. | 0.955 ± 0.008 | 0.911 ± 0.031 | 0.969 ± 0.003 | 0.840 ± 0.016 | 0.975 ± 0.006 |
| Isom. C9H10. | 0.850 ± 0.005 | 0.828 ± 0.126 | 0.897 ± 0.007 | 0.707 ± 0.040 | 0.875 ± 0.013 |
| JNK3 | 0.904 ± 0.004 | 0.785 ± 0.072 | 0.764 ± 0.069 | 0.683 ± 0.132 | 0.910 ± 0.021 |
| Median 1 | 0.340 ± 0.007 | 0.415 ± 0.001 | 0.379 ± 0.010 | 0.254 ± 0.017 | 0.357 ± 0.001 |
| Median 2 | 0.323 ± 0.005 | 0.408 ± 0.003 | 0.294 ± 0.007 | 0.226 ± 0.009 | 0.349 ± 0.001 |
| Mest. Sim. | 0.671 ± 0.021 | 0.930 ± 0.106 | 0.708 ± 0.057 | 0.480 ± 0.008 | 0.759 ± 0.023 |
| Osim. MPO | 0.866 ± 0.009 | 0.833 ± 0.011 | 0.860 ± 0.008 | 0.820 ± 0.003 | 0.856 ± 0.024 |
| Peri. MPO | 0.681 ± 0.017 | 0.651 ± 0.030 | 0.595 ± 0.014 | 0.556 ± 0.032 | 0.774 ± 0.006 |
| QED | 0.939 ± 0.001 | 0.947 ± 0.000 | 0.942 ± 0.000 | 0.938 ± 0.001 | 0.940 ± 0.002 |
| Rano. MPO | 0.820 ± 0.016 | 0.810 ± 0.011 | 0.819 ± 0.018 | 0.802 ± 0.009 | 0.839 ± 0.016 |
| Scaffold Hop | 0.576 ± 0.014 | 0.529 ± 0.020 | 0.615 ± 0.100 | 0.532 ± 0.014 | 0.541 ± 0.008 |
| Sita. MPO | 0.601 ± 0.011 | 0.474 ± 0.085 | 0.634 ± 0.039 | 0.348 ± 0.022 | 0.454 ± 0.074 |
| Thio. Redisc. | 0.584 ± 0.009 | 0.727 ± 0.089 | 0.583 ± 0.034 | 0.433 ± 0.033 | 0.647 ± 0.003 |
| Trog. Redisc. | 0.448 ± 0.017 | 0.756 ± 0.141 | 0.511 ± 0.054 | 0.322 ± 0.013 | 0.579 ± 0.002 |
| Zale. MPO | 0.486 ± 0.004 | 0.499 ± 0.025 | 0.552 ± 0.033 | 0.465 ± 0.017 | 0.529 ± 0.017 |
| Sum | 16.301 | 16.304 | 16.078 | 13.366 | 16.426 |

Table 15: Task-wise results of Table 4 (continued).

| Oracle Synthesis | REINVENT ✗ | MolGA ✗ | SynNet ✓ | SynGA ✓ | SynGBO ✓ |
|---|---|---|---|---|---|
| Albu. Sim. | 0.881 ± 0.016 | 0.928 ± 0.015 | 0.568 ± 0.033 | 0.649 ± 0.058 | 0.947 ± 0.024 |
| Amlo. MPO | 0.644 ± 0.019 | 0.740 ± 0.055 | 0.566 ± 0.006 | 0.573 ± 0.019 | 0.670 ± 0.088 |
| Cele. Redisc. | 0.717 ± 0.027 | 0.629 ± 0.062 | 0.439 ± 0.035 | 0.494 ± 0.063 | 0.856 ± 0.013 |
| Deco Hop | 0.662 ± 0.044 | 0.656 ± 0.013 | 0.635 ± 0.043 | 0.629 ± 0.014 | 0.831 ± 0.039 |
| DRD2 | 0.957 ± 0.007 | 0.950 ± 0.004 | 0.970 ± 0.006 | 0.976 ± 0.006 | 0.981 ± 0.010 |
| Fexo. MPO | 0.781 ± 0.013 | 0.835 ± 0.012 | 0.750 ± 0.016 | 0.773 ± 0.018 | 0.833 ± 0.018 |
| GSK3β | 0.885 ± 0.031 | 0.894 ± 0.025 | 0.713 ± 0.057 | 0.866 ± 0.072 | 0.924 ± 0.027 |
| Isom. C7H8. | 0.942 ± 0.012 | 0.926 ± 0.014 | 0.862 ± 0.004 | 0.840 ± 0.016 | 0.975 ± 0.006 |
| Isom. C9H10. | 0.838 ± 0.030 | 0.894 ± 0.005 | 0.657 ± 0.030 | 0.707 ± 0.040 | 0.875 ± 0.013 |
| JNK3 | 0.782 ± 0.029 | 0.835 ± 0.040 | 0.574 ± 0.103 | 0.683 ± 0.132 | 0.910 ± 0.021 |
| Median 1 | 0.363 ± 0.011 | 0.329 ± 0.006 | 0.236 ± 0.015 | 0.254 ± 0.017 | 0.357 ± 0.001 |
| Median 2 | 0.281 ± 0.002 | 0.284 ± 0.035 | 0.241 ± 0.007 | 0.226 ± 0.009 | 0.349 ± 0.001 |
| Mest. Sim. | 0.634 ± 0.042 | 0.762 ± 0.048 | 0.402 ± 0.017 | 0.480 ± 0.008 | 0.759 ± 0.023 |
| Osim. MPO | 0.834 ± 0.010 | 0.853 ± 0.005 | 0.793 ± 0.008 | 0.820 ± 0.003 | 0.856 ± 0.024 |
| Peri. MPO | 0.535 ± 0.015 | 0.610 ± 0.038 | 0.541 ± 0.021 | 0.556 ± 0.032 | 0.774 ± 0.006 |
| QED | 0.941 ± 0.000 | 0.941 ± 0.001 | 0.941 ± 0.001 | 0.938 ± 0.001 | 0.940 ± 0.002 |
| Rano. MPO | 0.770 ± 0.005 | 0.830 ± 0.010 | 0.749 ± 0.009 | 0.802 ± 0.009 | 0.839 ± 0.016 |
| Scaffold Hop | 0.551 ± 0.024 | 0.568 ± 0.017 | 0.506 ± 0.012 | 0.532 ± 0.014 | 0.541 ± 0.008 |
| Sita. MPO | 0.470 ± 0.041 | 0.677 ± 0.055 | 0.297 ± 0.033 | 0.348 ± 0.022 | 0.454 ± 0.074 |
| Thio. Redisc. | 0.544 ± 0.026 | 0.544 ± 0.067 | 0.397 ± 0.012 | 0.433 ± 0.033 | 0.647 ± 0.003 |
| Trog. Redisc. | 0.458 ± 0.018 | 0.487 ± 0.024 | 0.280 ± 0.006 | 0.322 ± 0.013 | 0.579 ± 0.002 |
| Zale. MPO | 0.533 ± 0.009 | 0.514 ± 0.033 | 0.493 ± 0.014 | 0.465 ± 0.017 | 0.529 ± 0.017 |
| Sum | 15.003 | 15.686 | 12.610 | 13.366 | 16.426 |

Table 16: Task-wise results of Table 4 (continued).

| Oracle
Synthesis | SynthesisNet
✓ | SynFlowNet
✓ | SynGA
✓ | SynGBO
✓ |
|---|---|---|---|---|
| Amlo. MPO | 0.608 | — | $0.573 \pm 0.019$ | $0.670 \pm 0.088$ |
| Cele. Redisc. | 0.582 | — | $0.494 \pm 0.063$ | $0.856 \pm 0.013$ |
| DRD2 | 0.960 | $0.885 \pm 0.027$ | $0.976 \pm 0.006$ | $0.981 \pm 0.010$ |
| Fexo. MPO | 0.791 | — | $0.773 \pm 0.018$ | $0.833 \pm 0.018$ |
| GSK3$\beta$ | 0.848 | $0.691 \pm 0.034$ | $0.866 \pm 0.072$ | $0.924 \pm 0.027$ |
| JNK3 | 0.639 | — | $0.683 \pm 0.132$ | $0.910 \pm 0.021$ |
| Median 1 | 0.305 | — | $0.254 \pm 0.017$ | $0.357 \pm 0.001$ |
| Median 2 | 0.257 | — | $0.226 \pm 0.009$ | $0.349 \pm 0.001$ |
| Osim. MPO | 0.810 | — | $0.820 \pm 0.003$ | $0.856 \pm 0.024$ |
| Peri. MPO | 0.524 | — | $0.556 \pm 0.032$ | $0.774 \pm 0.006$ |
| Rano. MPO | 0.741 | — | $0.802 \pm 0.009$ | $0.839 \pm 0.016$ |
| Sita. MPO | 0.313 | — | $0.348 \pm 0.022$ | $0.454 \pm 0.074$ |
| Zale. MPO | 0.528 | — | $0.465 \pm 0.017$ | $0.529 \pm 0.017$ |

Table 17: Extended metrics on the PMO benchmark: the mean score and AUC score for the top-$k$ molecules summed across tasks, as well as their average diversity. We report the mean and standard deviation over 5 seeds. The minor discrepancy in top-10 AUC scores with Table 4 is due to rounding. Here, we average the metrics per seed and round as a final step, whereas Table 4 computes the metrics task-wise, rounds, and then sums.

| | Metric | SynGA | SynGBO |
|---|---|---|---|
| Top-1 | Mean | $14.927 \pm 0.164$ | $17.460 \pm 0.142$ |
| Top-10 | Mean | $14.464 \pm 0.152$ | $17.195 \pm 0.115$ |
| | AUC | $13.369 \pm 0.175$ | $16.425 \pm 0.116$ |
| | Diversity | $0.514 \pm 0.008$ | $0.388 \pm 0.011$ |
| Top-100 | Mean | $13.699 \pm 0.199$ | $16.824 \pm 0.095$ |
| | AUC | $12.211 \pm 0.187$ | $15.856 \pm 0.113$ |
| | Diversity | $0.613 \pm 0.014$ | $0.462 \pm 0.010$ |

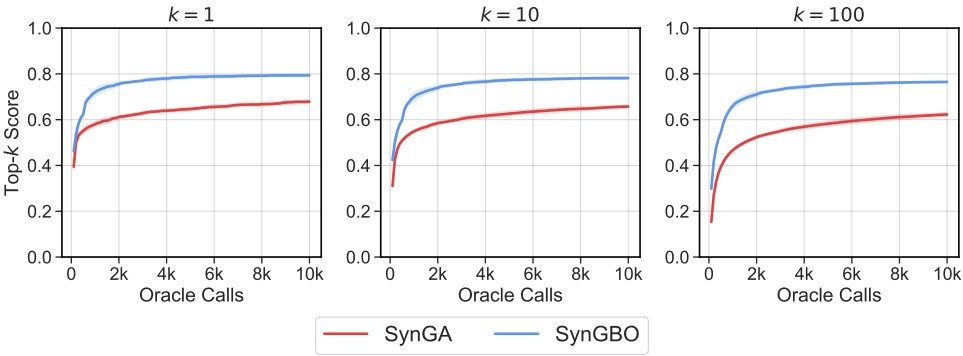

Figure 2: The mean score for the top-$k$ molecules plotted over the number of oracle calls consumed, and averaged over tasks. We plot the mean over 5 seeds.

Table 18: Vina docking scores of the top-100 diverse modes on LIT-PCBA dataset. Results for baselines are taken from Seo et al. (2025) and Shen et al. (2025) and use 64000 oracle calls whereas we use 16000. We report the mean and standard deviation over 4 seeds.

| Method | ADRB2 | ALDH1 | ESR_ago | ESR_ant | FEN1 |
|---|---|---|---|---|---|
| SynNet | $-8.03 \pm 0.26$ | $-8.81 \pm 0.21$ | $-8.88 \pm 0.13$ | $-8.52 \pm 0.16$ | $-6.36 \pm 0.09$ |
| BBAR | $-9.95 \pm 0.04$ | $-10.06 \pm 0.14$ | $-9.97 \pm 0.03$ | $-9.92 \pm 0.05$ | $-6.84 \pm 0.07$ |
| SynFlowNet | $-10.85 \pm 0.10$ | $-10.69 \pm 0.09$ | $-10.44 \pm 0.05$ | $-10.27 \pm 0.04$ | $-7.47 \pm 0.02$ |
| RGFN | $-9.84 \pm 0.21$ | $-9.93 \pm 0.11$ | $-9.99 \pm 0.11$ | $-9.72 \pm 0.14$ | $-6.92 \pm 0.06$ |
| RxnFlow | $-11.45 \pm 0.05$ | $-11.26 \pm 0.07$ | $-11.15 \pm 0.02$ | $-10.77 \pm 0.04$ | $-7.66 \pm 0.02$ |
| 3DSynthFlow | $-11.97 \pm 0.12$ | $-11.82 \pm 0.04$ | $\mathbf{-11.58} \pm 0.07$ | $-11.23 \pm 0.08$ | $\mathbf{-7.79} \pm 0.01$ |
| SynGA | $-11.98 \pm 0.27$ | $-11.97 \pm 0.04$ | $-11.23 \pm 0.11$ | $-11.07 \pm 0.20$ | $-7.61 \pm 0.06$ |
| SynGBO | $\mathbf{-12.16} \pm 0.42$ | $\mathbf{-12.36} \pm 0.04$ | $-11.45 \pm 0.05$ | $\mathbf{-11.68} \pm 0.21$ | $-7.71 \pm 0.12$ |

| Method | GBA | IDH1 | KAT2A | MAPK1 | MTORC1 |
|---|---|---|---|---|---|
| SynNet | $-7.60 \pm 0.09$ | $-8.74 \pm 0.08$ | $-7.64 \pm 0.38$ | $-7.33 \pm 0.14$ | $-9.30 \pm 0.45$ |
| BBAR | $-8.70 \pm 0.05$ | $-9.84 \pm 0.09$ | $-8.54 \pm 0.06$ | $-8.49 \pm 0.07$ | $-10.07 \pm 0.16$ |
| SynFlowNet | $-9.27 \pm 0.06$ | $-10.40 \pm 0.08$ | $-9.41 \pm 0.04$ | $-8.92 \pm 0.05$ | $-10.84 \pm 0.03$ |
| RGFN | $-8.48 \pm 0.06$ | $-9.49 \pm 0.13$ | $-8.53 \pm 0.11$ | $-8.22 \pm 0.15$ | $-9.89 \pm 0.06$ |
| RxnFlow | $-9.62 \pm 0.04$ | $-10.95 \pm 0.05$ | $-9.73 \pm 0.03$ | $-9.30 \pm 0.01$ | $-11.39 \pm 0.09$ |
| 3DSynthFlow | $-9.90 \pm 0.14$ | $-11.28 \pm 0.15$ | $\mathbf{-10.17} \pm 0.37$ | $-9.61 \pm 0.11$ | $-11.91 \pm 0.01$ |
| SynGA | $-9.65 \pm 0.16$ | $-11.27 \pm 0.21$ | $-10.08 \pm 0.11$ | $-9.38 \pm 0.20$ | $-12.04 \pm 0.45$ |
| SynGBO | $\mathbf{-10.33} \pm 0.09$ | $\mathbf{-11.76} \pm 0.28$ | $-10.08 \pm 0.10$ | $\mathbf{-9.64} \pm 0.09$ | $\mathbf{-12.28} \pm 0.26$ |

| Method | OPRK1 | PKM2 | PPARG | TP53 | VDR |
|---|---|---|---|---|---|
| SynNet | $-8.70 \pm 0.36$ | $-9.55 \pm 0.14$ | $-7.47 \pm 0.34$ | $-5.34 \pm 0.23$ | $-10.98 \pm 0.57$ |
| BBAR | $-9.84 \pm 0.10$ | $-11.39 \pm 0.08$ | $-8.69 \pm 0.10$ | $-7.05 \pm 0.09$ | $-11.07 \pm 0.04$ |
| SynFlowNet | $-10.34 \pm 0.07$ | $-11.98 \pm 0.12$ | $-9.40 \pm 0.05$ | $-7.90 \pm 0.10$ | $-11.62 \pm 0.13$ |
| RGFN | $-9.61 \pm 0.11$ | $-10.96 \pm 0.18$ | $-8.53 \pm 0.07$ | $-7.07 \pm 0.06$ | $-10.86 \pm 0.11$ |
| RxnFlow | $-10.84 \pm 0.03$ | $-12.53 \pm 0.02$ | $-9.73 \pm 0.02$ | $-8.09 \pm 0.06$ | $-12.30 \pm 0.07$ |
| 3DSynthFlow | $-11.26 \pm 0.41$ | $-13.36 \pm 0.03$ | $-10.00 \pm 0.04$ | $-8.41 \pm 0.17$ | $-12.98 \pm 0.10$ |
| SynGA | $-11.25 \pm 0.19$ | $-13.44 \pm 0.12$ | $-10.00 \pm 0.15$ | $-8.23 \pm 0.14$ | $-12.77 \pm 0.21$ |
| SynGBO | $\mathbf{-11.55} \pm 0.12$ | $\mathbf{-13.65} \pm 0.09$ | $\mathbf{-10.18} \pm 0.08$ | $\mathbf{-8.51} \pm 0.20$ | $-13.36 \pm 0.12$ |

Table 19: Ligand efficiency of the top-100 diverse modes on the LIT-PCBA dataset. Results for baselines are taken from Seo et al. (2025) and Shen et al. (2025) and use 64000 oracle calls whereas we use 16000. We report the mean and standard deviation over 4 seeds.

| Method | ADRB2 | ALDH1 | ESR_ago | ESR_ant | FEN1 |
|---|---|---|---|---|---|
| SynNet | $0.274 \pm 0.041$ | $0.272 \pm 0.006$ | $0.317 \pm 0.005$ | $0.289 \pm 0.020$ | $0.196 \pm 0.003$ |
| BBAR | $0.412 \pm 0.006$ | $0.401 \pm 0.008$ | $0.380 \pm 0.001$ | $0.387 \pm 0.003$ | $0.257 \pm 0.003$ |
| SynFlowNet | $0.401 \pm 0.006$ | $0.380 \pm 0.007$ | $0.361 \pm 0.003$ | $0.361 \pm 0.004$ | $0.247 \pm 0.004$ |
| RGFN | $0.393 \pm 0.005$ | $0.357 \pm 0.004$ | $0.346 \pm 0.002$ | $0.344 \pm 0.002$ | $0.241 \pm 0.001$ |
| RxnFlow | $0.412 \pm 0.005$ | $0.396 \pm 0.005$ | $0.375 \pm 0.002$ | $0.380 \pm 0.004$ | $0.246 \pm 0.001$ |
| 3DSynthFlow | $0.448 \pm 0.019$ | $0.395 \pm 0.006$ | $0.391 \pm 0.005$ | $0.398 \pm 0.016$ | $0.252 \pm 0.005$ |
| SynGA | $0.435 \pm 0.012$ | $0.411 \pm 0.010$ | $0.387 \pm 0.005$ | $0.385 \pm 0.006$ | $0.249 \pm 0.004$ |
| SynGBO | $\mathbf{0.486} \pm 0.019$ | $\mathbf{0.459} \pm 0.008$ | $\mathbf{0.422} \pm 0.008$ | $\mathbf{0.451} \pm 0.007$ | $\mathbf{0.285} \pm 0.006$ |

| Method | GBA | IDH1 | KAT2A | MAPK1 | MTORC1 |
|---|---|---|---|---|---|
| SynNet | $0.244 \pm 0.013$ | $0.281 \pm 0.016$ | $0.294 \pm 0.042$ | $0.226 \pm 0.004$ | $0.316 \pm 0.035$ |
| BBAR | $0.336 \pm 0.002$ | $0.382 \pm 0.005$ | $0.332 \pm 0.003$ | $0.320 \pm 0.002$ | $0.385 \pm 0.004$ |
| SynFlowNet | $0.330 \pm 0.004$ | $0.368 \pm 0.002$ | $0.327 \pm 0.003$ | $0.305 \pm 0.002$ | $0.368 \pm 0.002$ |
| RGFN | $0.310 \pm 0.002$ | $0.351 \pm 0.003$ | $0.310 \pm 0.003$ | $0.298 \pm 0.002$ | $0.346 \pm 0.004$ |
| RxnFlow | $0.327 \pm 0.004$ | $0.378 \pm 0.001$ | $0.330 \pm 0.001$ | $0.313 \pm 0.001$ | $0.370 \pm 0.001$ |
| 3DSynthFlow | $0.340 \pm 0.018$ | $0.388 \pm 0.007$ | $0.340 \pm 0.016$ | $0.322 \pm 0.009$ | $0.387 \pm 0.008$ |
| SynGA | $0.327 \pm 0.002$ | $0.401 \pm 0.009$ | $0.334 \pm 0.004$ | $0.328 \pm 0.003$ | $0.377 \pm 0.006$ |
| SynGBO | $\mathbf{0.390} \pm 0.006$ | $\mathbf{0.442} \pm 0.004$ | $\mathbf{0.378} \pm 0.005$ | $\mathbf{0.365} \pm 0.007$ | $\mathbf{0.421} \pm 0.008$ |

| Method | OPRK1 | PKM2 | PPARG | TP53 | VDR |
|---|---|---|---|---|---|
| SynNet | $0.298 \pm 0.039$ | $0.296 \pm 0.005$ | $0.253 \pm 0.031$ | $0.211 \pm 0.031$ | $0.359 \pm 0.015$ |
| BBAR | $0.370 \pm 0.006$ | $0.442 \pm 0.004$ | $0.326 \pm 0.007$ | $0.288 \pm 0.005$ | $0.409 \pm 0.002$ |
| SynFlowNet | $0.359 \pm 0.004$ | $0.427 \pm 0.003$ | $0.317 \pm 0.002$ | $0.287 \pm 0.008$ | $0.393 \pm 0.003$ |
| RGFN | $0.349 \pm 0.001$ | $0.405 \pm 0.002$ | $0.307 \pm 0.002$ | $0.271 \pm 0.001$ | $0.381 \pm 0.002$ |
| RxnFlow | $0.369 \pm 0.007$ | $0.436 \pm 0.005$ | $0.319 \pm 0.002$ | $0.289 \pm 0.003$ | $0.405 \pm 0.002$ |
| 3DSynthFlow | $0.389 \pm 0.006$ | $0.444 \pm 0.010$ | $0.321 \pm 0.005$ | $0.294 \pm 0.006$ | $0.416 \pm 0.009$ |
| SynGA | $0.383 \pm 0.006$ | $0.494 \pm 0.005$ | $0.329 \pm 0.005$ | $0.299 \pm 0.011$ | $0.415 \pm 0.010$ |
| SynGBO | $\mathbf{0.459} \pm 0.013$ | $\mathbf{0.543} \pm 0.005$ | $\mathbf{0.363} \pm 0.006$ | $\mathbf{0.336} \pm 0.008$ | $\mathbf{0.465} \pm 0.009$ |

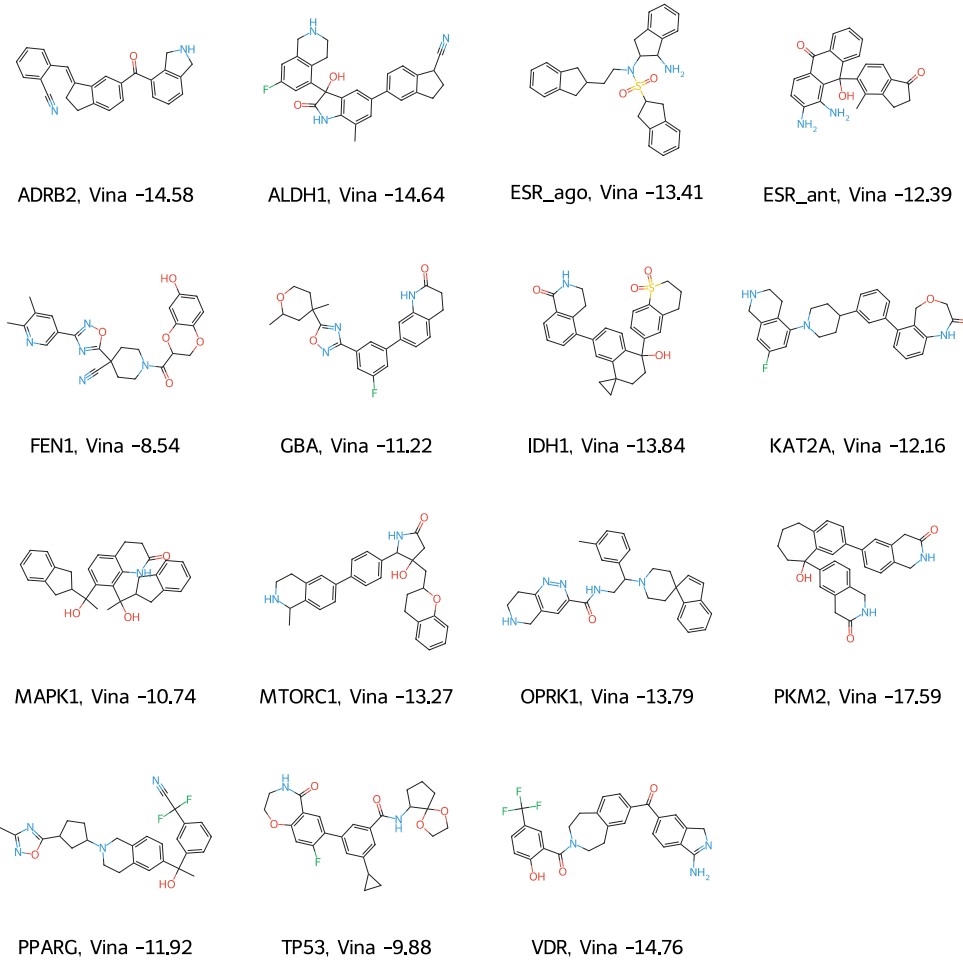

Figure 3: An example ligand proposed by SynGA for each receptor.

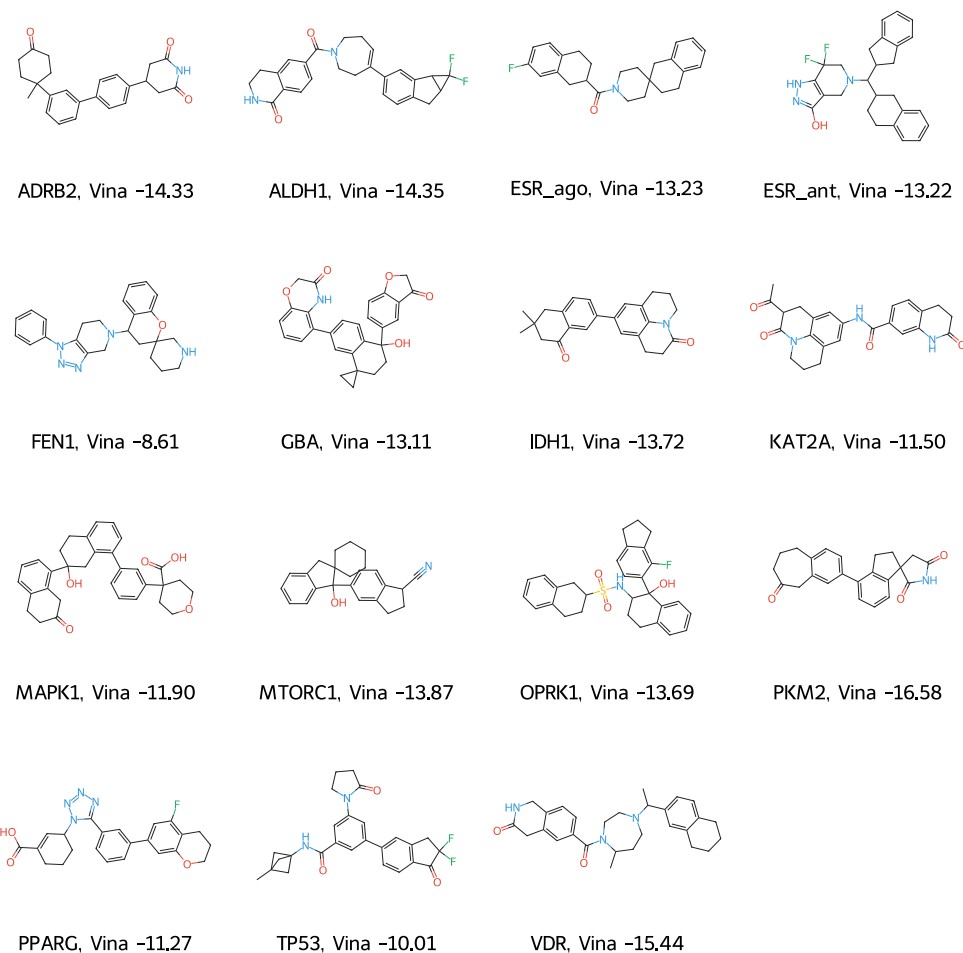

Figure 4: An example ligand proposed by SynGBO for each receptor.

