# OpenReview forum: "A Genetic Algorithm for Navigating Synthesizable Molecular Spaces"
_ICLR.cc/2026/Conference — ICLR 2026 Poster_

### Official Review · Reviewer_taeo · 2025-10-19

**Soundness:** 3
**Presentation:** 4
**Contribution:** 3
**Rating:** 6
**Confidence:** 2

**Summary:**

This paper presents SynGA, a synthesis-constrained genetic algorithm that operates directly on molecular synthesis routes. By using custom crossover and mutation operators, it ensures all generated molecules are synthesizable and can be efficiently optimized for both analog search and property objectives in 2D and 3D. The method is further enhanced with a lightweight ML-based building block filter, and its variant SynGBO achieves state-of-the-art performance on benchmark property optimization and docking tasks. Overall, the work demonstrates a flexible and effective approach for synthesis-aware molecular design.

**Strengths:**

1. The idea of measuring synthesizability directly from synthesis trees is insightful.

2. The paper is well-structured and clearly organized, making the methodology and results easy to follow.

3. The experiments are solid and thorough, and the results look pretty convincing.

**Weaknesses:**

1. Some parts of the method feel a bit engineering-heavy, and the paper is missing some discussion. For example, when they say “we couple the NAM with a more powerful predictor that filters the samples post-hoc from SynGA”, it’s not clear how much the extra predictor actually contributes beyond the NAM. Also the five mutation operations.
2. The algorithm relies on expert-defined SMARTS strings (templates), and here they only use 91 reactions templates. If more templates were used, like those extracted in Retro*, the algorithm might face more pressure. That said, this may not be a big issue since related work also uses a similar number of templates.

**Questions:**

1. In line 166, $R(M(S1),M(S2)) \not= \emptyset$ is a bit hard to follow, though the meaning can still be understood.

2. I wonder what would happen if the building block filtering considered all the building-block molecules used in multiple feasible synthesis paths for a target, instead of just a path.

3. The table reports averages over 5 seeds, but the variance is not shown.

4. In Table 2, it would be interesting to know how the results of SynGA (MLP) would change if its runtime were also 80m, matching SynFormer, and how its performance varies under different computational budgets.

---

> ### Author Response · Authors · 2025-11-21
>
> **Thank you for the kind words and thoughtful feedback\!**
>
> **On the engineering of SynGA**
>
> We ablate various components of our method in the Appendix. To summarize, in Appendix A.4, we ablate SynGA’s mutation operations and find that it is robust across multiple settings. In Appendix C.1, we ablate the impact of the NAM and GP predictor on small datasets and find their coupling outperforms each component in isolation. In particular, the performance gain for the NAM-GP over the NAM-only setup is greater for properties that are less additive in nature.
>
> **On the template library size**
>
> Scaling to large template libraries would certainly be interesting for future work. However, a potential concern with using retrosynthesis-style libraries is their robustness, since they are computationally mined from patents as opposed to expert-curated (like those used in SynGA and other related work). Likely, one would have to address this by incorporating a feasibility or relevance model in the search, as is done in retrosynthesis.
>
> **(Q2) Building block filtering and multiple feasible paths**
>
> Conceptually, our block filtering approach can handle different feasible paths for a target using different blocks. The main barrier is that we very rarely stumble upon such cases in practice (e.g., through route sampling or genetic operators) due to the large size of synthesis route space. To address this, one might have to directly search for alternative paths using retrosynthesis tools, but this may incur significant costs if applied naively. Apologies in advance if we have misunderstood your question\!
>
> **(Q3) Table 4 variances**
>
> We have included this in Appendix C.5 (Tables 13-16).
>
> **(Q4) SynGA analog search on a limited budget**
>
> Unfortunately, it is difficult to tie runtime exactly since our machines are experiencing high usage during this rebuttal period, which makes our code run slower than before. Instead, we run SynGA for 3.3k oracle calls to roughly match the 3x slowdown of SynGA to SynFormer. The initial population size is also scaled down proportionally and all other hyperparameters are unchanged. We find that even with a third of the budget, SynGA exhibits strong (though slightly degraded) performance on analog search.
>
> | Method | Valid | RR | Morgan | Scaffold | Gobbi |
> | :---- | :---- | :---- | :---- | :---- | :---- |
> | SynFormer | 0.998 | 0.190 | 0.668 | 0.667 | 0.635 |
> | SynGA \+ MLP (10k) | 1.000 | 0.196 | 0.711 | 0.694 | 0.623 |
> | SynGA \+ MLP (3.3k) | 1.000 | 0.175 | 0.690 | 0.675 | 0.609  |

---

### Official Review · Reviewer_KFNZ · 2025-10-29

**Soundness:** 3
**Presentation:** 3
**Contribution:** 3
**Rating:** 4
**Confidence:** 3

**Summary:**

This paper proposes SynGA, a novel genetic algorithm that directly models and manipulates synthesis routes to efficiently explore the combinatorially vast space of synthesizable molecular pathways. The method demonstrates promising performance across multiple benchmarks, including docking and property optimization tasks.

**Strengths:**

- The proposed method, SynGA, enables direct control of synthesis routes without ML tuning, while ensuring synthetic feasibility.
- The paper is well structured and clearly explains the methodology for designing synthesis routes directly.
- SynGA and SynBO show good performance across various benchmarks.

**Weaknesses:**

- Although one of the key advantages of SynGA is its ML-free design, its performance drops without ML-guided building block filtering. In practice, meaningful results appear only when combined with ML filtering. Moreover, In both PMO and docking tasks, the performance improvement is achieved only when using the computationally expensive BO algorithm, which weakens the claim of being ML-free.
- While the method performs well across benchmarks, results are reported for only a few targets.

**Questions:**

- One of SynGA’s strengths is its ML-free structure, but its performance drops when ML-guided building block filtering is not used (probably because the search space is too large). While the process of constructing synthesis routes directly through custom genetic operators is novel, the performance seems to rely heavily on the filtering method. Could additional datasets such as Enamine[1] or ZINC250[2] be included for comparison? Evaluating the effect of the proposed approach (especially the ML-guided building block filtering) across different datasets in the synthesizable analog search task would help demonstrate the generality of SynGA.

- In molecular design, diversity can be as important as chemical validity [3]. Would it be possible to evaluate diversity in the analog search experiments?

- Could you clarify why only a subset of tasks is reported in Table 3? According to [4], the LIT-PCBA dataset includes 15 protein targets.

- Similarly, Table 5 seems to report results for only a few targets. Prior works such as RXNFLOW[5] and 3DSynthFlow[6] include results not only for ALDH1 and ESR_ant, but also for ADRB2, ESR_ago, and FEN1. Could you clarify the selection criteria for the reported targets? If there was no specific reason for selecting only these three, results for additional targets—as in previous studies—would make the evaluation more comprehensive.

- Could optimization curves, similar to those shown in previous PMO papers (e.g., PMO[7], MOLLEO[8], DyMol[9]), be included for additional analysis? These curves might help better illustrate the sample efficiency of SynGA and SynGBO.

[1] Lee, Seul, et al. "Rethinking Molecule Synthesizability with Chain-of-Reaction." arXiv preprint arXiv:2509.16084 (2025).

[2] Irwin, John J., et al. "ZINC: a free tool to discover chemistry for biology." Journal of chemical information and modeling 52.7 (2012): 1757-1768.

[3] Renz, Philipp, Sohvi Luukkonen, and Günter Klambauer. "Diverse hits in de novo molecule design: Diversity-based comparison of goal-directed generators." Journal of Chemical Information and Modeling 64.15 (2024): 5756-5761.

[4] Luo, Shitong, et al. "Projecting molecules into synthesizable chemical spaces, 2024." URL https://arxiv.org/abs/2406 4628.

[5] Seo, Seonghwan, et al. "Generative flows on synthetic pathway for drug design." arXiv preprint arXiv:2410.04542 (2024).

[6] Shen, Tony, et al. "Compositional Flows for 3D Molecule and Synthesis Pathway Co-design." arXiv preprint arXiv:2504.08051 (2025).

[7] Gao, Wenhao, et al. "Sample efficiency matters: a benchmark for practical molecular optimization." Advances in neural information processing systems 35 (2022): 21342-21357.

[8] Wang, Haorui, et al. "Efficient evolutionary search over chemical space with large language models." arXiv preprint arXiv:2406.16976 (2024).

[9] Shin, Dong-Hee, et al. "Dynamic many-objective molecular optimization: Unfolding complexity with objective decomposition and progressive optimization." Proceedings of the Thirty-Third International Joint Conference on Artificial Intelligence (IJCAI). 2024.

---

> ### Author Response · Authors · 2025-11-21
>
> **Thank you for the kind words and thoughtful feedback\!**
>
> **(Q1) SynGA and ML**
>
> Although SynGA is not SOTA, we still believe it achieves notable results, beating even some ML-based methods. For example:
>
> - SynGA with an ML-free similarity filter outperforms all baselines except SynFormer on ChEMBL.
> - On PMO, SynGA performs comparably to SynthesisNet and outperforms all other synthesis-aware baselines. We note that SynNet itself was the top synthesis baseline at the time of publication of PMO, and emphasize the difficulty of competing with synthesis-agnostic baselines that operate on much larger and more granular search spaces.
> - On our docking benchmark, SynGA outperforms all baselines except 3DSynthFlow.
>
> However, we clarify that our main position is that while we expect classical GAs (e.g., MolGA, SynGA) to be highly performant, we believe the *best* results come from coupling GAs with ML (e.g., f-RAG, GPBO, SynGBO). Our experiments showcase how complementing SynGA with relatively small ML components produces SOTA synthesizable results. The ML-free and lightweight nature of SynGA is what allows it to be flexibly hybridized, such as in SynGBO where we can afford to call SynGA 100x more steps as a subroutine.
>
> **(Q1-2) Extended analog search experiments**
>
> We add extended metrics (Appendix Table 8\) to our revised manuscript using 1k molecules from the Enamine Discovery Diversity Set (DDS-10) and ZINC as additional test sets. Additionally, we report the average diversity of the top 5 analogs for each query. We compare against the top two baselines, ChemProjector and SynFormer.
>
> In summary, SynGA consistently outperforms ChemProjector, but SynFormer outperforms SynGA on DDS-10 and ZINC. The improvement of SynFormer is modest for similarity metrics (\~0.03) but considerably higher for reconstruction (\>10%). We attribute this to DDS-10 and ZINC being much more similar to the search space of SynFormer; many molecules are “reachable” or highly similar to “reachable” molecules. Since SynFormer has been trained extensively for synthesizable reconstruction (i.e., retrosynthesis) within its search space, in such cases, it outperforms SynGA, which has only been trained on a block relevance task. On more out-of-distribution datasets like ChEMBL or ML-designed molecules (Section 4.2.3), we find that SynGA performs better. Meanwhile, the diversity metrics are generally inversely correlated to the similarity metrics, which is expected since analogs that are more similar to the query are usually more similar to each other.
>
> We have updated Section 4.2.2 and Appendix B.5 (new) with further analysis and discussion. As mentioned in our outlook (Appendix D), we believe it is promising to explore the hybridization of SynGA with ammortized methods like SynFormer to make up for each method’s weaknesses.
>
> **(Q3-4) LIT-PCBA benchmarks**
>
> We originally subset to the 3 receptors arbitrarily to leave bandwidth to pursue a wider variety of experiments. We are unable to obtain results on all 15 LIT-PCBA receptors during the rebuttal period, as docking is particularly costly. To make our results more comprehensive, we add 2 more random targets (MAPK1, PPARG) to our docking benchmark (Table 5 and Appendix C.6) in our revised manuscript. We prioritize Table 5 since Table 3 is largely downstream of and supported by prior analog search experiments. Our additional results are aligned with our prior metrics and discussion on ALDH1, ESR\_ant, and TP53.
>
> **(Q5) PMO optimization curves**
>
> We have added optimization curves of SynGA and SynGBO to Appendix C.5 (Figure 2). Unfortunately, we are unable to overlay the optimization curves of any baselines, since we use numbers from prior work and the necessary data is unavailable. We will release our raw data upon publication, so that SynGA and SynGBO may be easily incorporated into future plots.

---

### Official Review · Reviewer_SYG6 · 2025-11-01

**Soundness:** 2
**Presentation:** 3
**Contribution:** 3
**Rating:** 6
**Confidence:** 4

**Summary:**

The authors propose a template-based genetic algorithm for molecular generation. Furthermore, they propose a model-based fragment filtering step that improves the performance of the method. The approach is evaluated on multiple benchmarks.

**Strengths:**

- The approach is sound, and presents an interesting contribution.
- The area of focus (synthesizability-constrained generation) is of high importance.
- The paper is clearly written and, for the most part, easy to follow.
- The experiments are comprehensive and measure the performance of the proposed approach in multiple relevant tasks.

**Weaknesses:**

- The compared baselines are not standardized in terms of reaction set, fragments/synthons, and the number of allowed reactions used. The choice of chemical vocabulary greatly affects the performance. It is not clear to what extent reported performance improvement compared to baselines is due to a better algorithm, or the vocabulary.

**Questions:**

- I am familiar with an earlier version of the paper submitted to a previous conference. I noticed that the performance for 3DSynthFlow in Table 5 has decreased compared to the previous version. This is a substantial change, since previously this was the best performing method, outperforming SynGA and within the standard deviation of SynGBO. As a result, the change affects the conclusions to some extent. Can authors clarify why the results were changed?
- More of a suggestion, but in my opinion it would make more sense to put the paragraph describing SynGBO in a separate subsection.

---

> ### Author Response · Authors · 2025-11-21
>
> **Thank you for the kind words and thoughtful feedback\!**
>
> We have moved SynGBO into a separate subsection (new Section 3.4) in our revised manuscript.
>
> **Standardizing chemical vocabularies**
>
> Unfortunately, it is difficult to standardize all search spaces exactly since we take results from prior papers and baselines have differing algorithmic requirements. For example, 3DSynthFlow operates on synthons but not building blocks, and SynFlowNet’s performance seems to degrade when more blocks are used (Shen et al., 2025). To take a step towards addressing this, we are actively working on an ablation that tries to match the 3DSynthFlow search space and will send an update when completed.
>
> **(Q1) 3DSynthFlow Table 5**
>
> An earlier version of this paper was submitted when 3DSynthFlow was a preprint (v1 on their arXiv). We have since updated our docking tables to reflect the published ICML version of 3DSynthFlow. A changelog is not included in their paper, but running a pdfdiff between the two versions shows differences in their modelling choices and hyperparameters. Out of the 3 receptors evaluated, ALDH1 (+0.43) seems to have the biggest mean change, though still within a standard deviation of their prior result, whereas ESR\_ant (+0.02) and TP53 (+0.08) are relatively minor. We have added a brief note to the Table 5 caption clarifying this.

---

> > ### Author Response · Authors · 2025-11-27
> >
> > **Apologies for the delay\!**
> >
> > We make our best attempt at matching the search space of the top-performing baseline 3DSynthFlow on the docking benchmark. We note that 3DSynthFlow operates on brick and linker synthons (induced by the templates on building blocks) as opposed to blocks themselves, which may limit the types of templates it can use.
> >
> > According to their paper, 3DSynthFlow uses 1.2M blocks processed from Enamine’s comprehensive catalog v2024.06.10. However, the catalog has since grown to 1.6M blocks and we are only able to access the current-day version. Thus, we reprocess the current catalog and randomly downsample to 1.2M blocks. In parallel, we had also reached out to the authors of 3DSynthFlow. They suggested we use a different 300k block set from Enamine’s in-stock set, stating that there may be a discrepancy in their paper. Thus, we will use both the catalog (1.2M) and stock (300k) sets in our ablation. In both cases, we use the same 38 bimolecular reaction templates, 2 maximum reaction steps, and 40 maximum atoms.
> >
> > We report our results on two receptors, ESR\_ant and TP53, due to limited budget. All hyperparameters are kept the same, except we filter the top 10k blocks when using the 1.2M catalog set to accommodate for the larger space. Only the top 3 baselines are shown for conciseness:
> >
> > | Method | Search Space | ESR\_ant | TP53 |
> > | :---- | :---- | :---- | :---- |
> > | SynFlowNet |  | −10.27 ± 0.04 | −7.90 ± 0.10 |
> > | RxnFlow |  | −10.77 ± 0.04 | −8.09 ± 0.06 |
> > | 3DSynthFlow |  | −11.23 ± 0.08 | −8.41 ± 0.17 |
> > | SynGA | Ours | −11.07 ± 0.20 | −8.23 ± 0.14 |
> > | SynGA | 3DSynthFlow (catalog, 1.2M) | −10.92 ± 0.15 | −8.17 ± 0.25 |
> > | SynGA | 3DSynthFlow (stock, 300k) | −10.89 ± 0.13 | −8.24 ± 0.13 |
> > | SynGBO | Ours | −11.68 ± 0.21 | −8.51 ± 0.20 |
> > | SynGBO | 3DSynthFlow (catalog, 1.2M) | −11.39 ± 0.12 | −8.72 ± 0.33 |
> > | SynGBO | 3DSynthFlow (stock, 300k) | −11.53 ± 0.04 | −8.45 ± 0.41 |
> >
> > In general, we find changing the building block set has a minor impact on the numeric scores. Our general conclusions remain unchanged: using a quarter of the oracle calls, SynGBO outperforms or competes with 3DSynthFlow, and SynGA outperforms or competes with all other baselines.

---

### Official Review · Reviewer_fXNd · 2025-11-09

**Soundness:** 4
**Presentation:** 3
**Contribution:** 3
**Rating:** 8
**Confidence:** 5

**Summary:**

very solid paper proposing a new genetic algorithm for molecular design combined with ML based building block selection; with convincing evaluation and a good contribution to the field.

**Strengths:**

- clear description of the work
- good, well-motivated method contribution
- proper evaluation in line with prior work

**Weaknesses:**

No major weaknesses.

if I had to really pick some, I'd say the language of the paper is maybe in parts unnecessarily formal, and the contribution isn't paradigm-shifting (but nevertheless very solid!).

**Questions:**

- do the authors think that number of oracle calls is really important in practice?

---

> ### Author Response · Authors · 2025-11-21
>
> **Thank you for the kind words and thoughtful feedback\!**
>
> We believe that sample efficiency is an important practical consideration, since some property oracles of interest are costly to evaluate, e.g., properties involving expensive simulations or wet lab experiments. In silico, a more sample-efficient algorithm can not only improve the throughput and efficiency of workflows but also allow practitioners to pursue higher-fidelity computational oracles.

---

### Meta-Review · Area_Chair_ZUKx · 2025-12-12

**Summary:**

This paper presents a new genetic algorithm (GA) specifically designed to propose custom, synthesizable operations on SMARTS strings for molecular design tasks called SynGA. The method can use machine learning surrogates for screening the proposed sequences using ideas from Bayesian optimization (BO), resulting in a method called SynGBO.

The major strengths and weaknesses pointed out by the reviewers are as follows.

**Strengths**:

- Clearly presented method and clearly written paper
- Clear contribution (operations remain in the space of synthesizable chemicals) and practical utility of the method
- Good empirical evaluation

**Weaknesses**:

- Some baselines have non-standard design spaces or have shown better performance in previous versions of the work (mostly rectified)
- Some relevant benchmark experiments are missing or require more details (partially rectified)

Overall I believe the reviewer concerns have been largely addressed in the rebuttal, with the rest being achievable by the camera ready deadline, hence I am suggesting this paper be accepted.

Please make sure the additional experiments requested are in the camera ready paper.

**Reviewer Concerns:**

Most requests centred around additional experiments, ablations or experimental detail. These were largely addressed through discussion, or where possible by the authors as additional experiments prepared in the discussion period, or pointing to existing appendices or clarifying results.

Some of the additional experiments requested by reviewer KFNZ were not possible to implement in the time period (but two were), and some of the timing-based results (tied runtime for the MLP variant) requested by reviewer taeo were not possible to implement precisely due to resource constraints. I believe these can be integrated into the camera ready version of the paper.

**Reviewer Scores:**

fXNd -- unchanged 8

SYG6 -- could have moved from 6 to 8

KFNZ -- could have moved from 4 to 6

taeo -- I believe would have possibly remained at 6 since some of the criticisms are hard to address (engineering heavy method, reaction representations), however a good faith attempt was made at others.

---

### Decision · Program_Chairs · 2026-01-26

Accept (Poster)